# RETHINKING IDENTITY IN KNOWLEDGE GRAPH EMBEDDING

## ABSTRACT

Knowledge Graph Embedding (KGE) is a common method to complete real-world Knowledge Graphs (KGs) by learning the embeddings of entities and relations. Beyond specific KGE models, previous work proposes a general framework based on group. A group has a special element **identity** that uniquely corresponds to the relation *identity* in KGs, which implies that *identity* should be represented uniquely. However, we find that this uniqueness cannot be modeled by bilinear based models, revealing the inconsistency between the framework and models. To this end, we propose a solution named Unit Ball Bilinear Model (UniBi). In addition to theoretical superiority, it has greater interpretability and improves performance by preventing ineffective learning with the least constraints. Experiments demonstrate that UniBi models the uniqueness and verify its interpretability and performance.

## 1 INTRODUCTION

Knowledge Graphs (KGs) store human knowledge in the form of triple $(h, r, t)$, which represents a relation $r$ between a head entity $h$ and a tail entity $t$ (Ji et al., 2021). KGs benefit a lots of downstream tasks and applications, e.g., recommender system (Zhang et al., 2016), dialogue system (He et al., 2017) and question answering (Mohammed et al., 2018). Since actual KGs are usually incomplete, researchers are interested in predicting missing links to complete them. As a common solution, Knowledge Graph Embedding (KGE) completes KGs by learning low-dimensional representations of entities and relations.

Beyond the great advances in specific KGE models (Trouillon et al., 2016; Hitchcock, 1927; Chami et al., 2020; Liu et al., 2017; Nickel et al., 2011; Bordes et al., 2013), several works also attempt to unify these models with general frameworks, such as promising ones based on group (Yang et al., 2020; Xu & Li, 2019; Ebisu & Ichise, 2018). Group is an abstraction of an operations on a set, like addition on integer. Just like such case has a special number 0, each group has a unique element **identity**. From the perspective of group, such element requires that its correspondence in KGs, *identity* relation , should be represented uniquely. However, we find that such uniqueness cannot be modeled by bilinear based models, which reveals the inconsistency between the framework and models.

To present the problem more clearly, we first need to introduce some notation. A model with a score function $s(h, r, t)$ can model the uniqueness of *identity* means that $\forall h \neq t, s(h, r, h) > s(h, r, t)$ holds if and only if $r$ is *identity* and its universal representation is unique. In addition, the score function $s(\cdot)$ of bilinear based model is $\mathbf{h}^\top \mathbf{R} \mathbf{t}$, where $\mathbf{h}, \mathbf{R}, \mathbf{t}$ are the representations of $h$, $r$, and $t$.

In terms of such uniqueness, bilinear based models have two flaws. On the one hand, Fig. 1(a) demonstrates $\mathbf{e}_1^\top \mathbf{I} \mathbf{e}_1 < \mathbf{e}_1^\top \mathbf{I} \mathbf{e}_2$, which means that the relation matrices per se do not model *identity* perfectly. On the other hand, Fig. 1(b) shows even if a matrix, e.g. $\mathbf{I}$, does. Its scaled one $k\mathbf{I}$ can also model *identity* and thus breaks the uniqueness.

Obviously, modeling this property requires both entities and relations to be restricted, which reduces expressiveness. To avoid this side effect, we make the cost negligible by minimizing the constraints, one per entity or relation, while modeling the desired property. To be specific, we normalize the vectors of the entities and the spectral radius of the matrices of the relations to 1. Since the model captures entities in a unit ball as shown in Fig. 1(c), we name it Unit Ball Bilinear Model (UniBi)

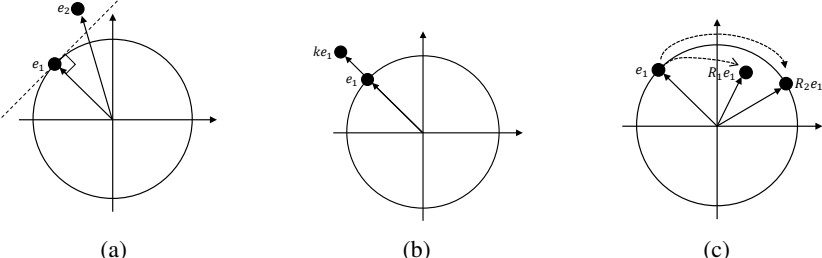

(a)         (b)         (c)

Figure 1: The flaws of bilinear based models and our solution in terms of modeling the uniqueness of *identity*. (a) Identity matrix fails to model *identity*. (b) Scaled identity matrix could also model *identity*. (c) An illustration of UniBi. All entities are embedded in the unit sphere and stay in the unit ball after relation specific transformations.

In addition to the theoretical superiority, UniBi is more powerful and interpretable since modeling identity uniquely requires normalizing the scales that barely contain any useful knowledge. On the one hand, scale normalization prevents ineffective learning on scales and makes UniBi focus more on useful knowledge. On the other hand, it reveals the connection between the relative ratio of singular values and the complexity of relations.

Experiments verify that UniBi models *identity* uniquely with improvement on performance and interpretability. Therefore, UniBi reconciles the framework and bilinear based models in terms of *identity* and paves the way for further studies of both.

## 2 PRELIMINARIES

### 2.1 BACKGROUND

A Knowledge Graph $\mathcal{K}$ is a set that contains the facts about sets of entities $\mathcal{E}$ and relations $\mathcal{R}$. Each fact is stored by a triple $(e_i, r_j, e_k) \in \mathcal{E} \times \mathcal{R} \times \mathcal{E}$ where $e_i$ and $r_j$ denote the $i$-th entity and the $j$-th relation, respectively.

KGE aims to predict the missing links in $\mathcal{K}$ by learning embeddings for each entity and relation via a score function $s : \mathcal{E} \times \mathcal{R} \times \mathcal{E} \to \mathbb{R}$. To verify the performance of a KGE method, $\mathcal{K}$ is first divided into $\mathcal{K}_{train}$ and $\mathcal{K}_{test}$. Then, the method is trained on $\mathcal{K}_{train}$ to learn the embeddings $\mathbf{e}$ and $\mathbf{r}$ (or $\mathbf{R}$) for each entity $e \in \mathcal{E}$ and relation $r \in \mathcal{R}$. Finally, the model is expected to give a higher rank if $(e_i, h_j, e_k) \in \mathcal{K}$ while a lower rank if $(e_i, h_j, e_k) \notin \mathcal{K}$, for each query $(e_i, h_j, ?)$ from $\mathcal{K}_{test}$ and the candidate entity $e_k \in \mathcal{E}$.

In addition to the above tail prediction, models are also required to test on head prediction conversely. We transform it to tail prediction by introducing reciprocal relations following Lacroix et al. (2018).

Group is the abstraction of an operation on a set. For example, additive group of integers is consists of addition and all integers, and its **identity** element is 0. Since we do not expand on this concept in the main text, we put the formal definition in Appendix B. To avoid confusion, we use bold to represent the element **identity** while italics to denote *identity* and other relations. And we gives the definition of modeling *identity* uniquely in the following.

**Definition 1.** *A KGE model can uniquely model identity means $s(h, r, h) > s(h, r, t), \forall h, t \in \mathcal{E}, h \neq t$ holds if and only if $r$ is identity and its universal representation, i.e., it works in any KG, is unique.*

### 2.2 OTHER NOTATIONS

We utilize $\hat{\mathbb{E}}$ and $\hat{\mathbb{R}}$ to denote the set of all possible representations of entities and relations. And we use $\mathbf{e} \in \hat{\mathbb{E}}$ and $\mathbf{R} \in \hat{\mathbb{R}}$ to denote the embedding vector of the entity $e$ and the transformation matrix specific to the relation $\mathbf{R}$. Furthermore, we use $\| \cdot \|$ to denote the L2 norm of the vectors, $\| \cdot \|_F$ and $\rho(\cdot)$ to represent the Frobenius norm and the spectral radius of a matrix.

In this paper, we focus on $n$-dimensional real space $\mathbb{R}^n$, which means $\hat{\mathbb{E}} \subseteq \mathbb{R}^n$ and $\hat{\mathbb{R}} \subseteq \mathbb{R}^{n \times n}$. We also consider real vector spaces whose vectors are complex $\mathbb{C}^n$ or hypercomplex space $\mathbb{H}^n$, since they are isomorphic to $\mathbb{R}^{2n}$ or $\mathbb{R}^{4n}$.

## 3 RELATED WORK

Previous work of KGE can be roughly divided into the following three categories: distance, bilinear and others.

Distance based models choose Euclidean distance for their score functions. TransE (Bordes et al., 2013) inspired by Word2Vec (Mikolov et al., 2013) in Natural Language Processing proposes the first distance based model, which uses translation as the linear transformation $s(h, r, t) = -\|\mathbf{h} + \mathbf{r} - \mathbf{t}\|$. TransH (Lin et al., 2015) and TransR (Lin et al., 2015) find that TransE difficult to handle complex relations and thus apply linear projections before translation. Apart from translation, RotatE (Sun et al., 2019) first introduces rotation as the transformation. RotE (Chami et al., 2020) further combines translation and rotation. Some works also introduce hyperbolic spaces (Balazevic et al., 2019b; Chami et al., 2020; Wang et al., 2021). Apparently, distance based models are capable of uniquely modeling *identity*, since the distance between a vector and itself is $0$[1].

In contrast, bilinear based models have score functions in the bilinear form $s(h, r, t) = \mathbf{h}^\top \mathbf{R} \mathbf{t}$. RESCAL (Nickel et al., 2011) is the first bilinear based model whose relation matrices are unconstrained. Although RESCAL is expressive, it contains too many parameters and tends to overfitting. DistMult (Yang et al., 2015) simplifies these matrices into diagonal ones. ComplEx (Trouillon et al., 2016) further introduces complex values to model the skew-symmetry pattern. Analogy (Liu et al., 2017) uses block-diagonal to model the analogical pattern and subsumes DistMult, ComplEx, and HolE (Nickel et al., 2016). Moreover, QuatE (Zhang et al., 2019) extends complex values to quaternion and GeomE (Xu et al., 2020) utilizes geometric algebra to subsume all these models. However, these studies have not noticed that bilinear based models fail to uniquely model *identity*.

Corresponding to *identity*, **identity** is a special element in a group which is first introduced by Dihedral (Xu & Li, 2019) and HolE (Nickel et al., 2016). Based on group, NagE (Yang et al., 2020) proposes a general framework to incorporate previous methods. Although they believe that the matrix representation for *identity* is $\mathbf{I}$, they ignore the fact that *identity* may not be uniquely modeled by specific models.

In addition, other works using black-box networks (Dettmers et al., 2018; Nguyen et al., 2018; Yao et al., 2019; Schlichtkrull et al., 2018; Zhang et al., 2020b) or additional information (An et al., 2018; Ren et al., 2016) are beyond the scope of this paper.

## 4 METHOD

In this section, we first discuss the relationship between the **identity** element in groups and the *identity* relation in KGs, and give the condition for bilinear based models to model *identity* uniquely in Section 4.1. We then propose a model named UniBi that satisfies it with least constraint in Section 4.2 and an efficient modeling for UniBi in Section 4.3. In addition, we also discuss its improvement on performance and interpretability via scale normalization in Section 4.4.

### 4.1 IDENTITY AND *IDENTITY*

In previous work, a framework based on group has been proposed (Yang et al., 2020). It states that all relations should be embedded as elements of a group. Groups have a special element **identity**, which corresponds to the *identity* relation in KGs. This correspondence implies that *identity* should have a unique representation.

Counter-intuitively, we find that in fact bilinear based models fail to model this uniqueness. We demonstrate two cases in which bilinear based models violate it. On the one hand, Fig. 1(a) demonstrates $\mathbf{e}_1^\top \mathbf{I} \mathbf{e}_1 < \mathbf{e}_1^\top \mathbf{I} \mathbf{e}_2$, which means that the matrix of a relation per se is not guaranteed for modeling *identity*. On the other hand, Fig. 1(b) shows even if a matrix, e.g., $\mathbf{I}$, does. Its scaled one

---

[1]We also prove this more rigorously in Appendix C.

$k\mathbf{I}$ where $k > 0, k \neq 1$ can also model *identity*, which contradicts the quantification of uniqueness. Therefore, we give a formal definition based on definition 1 as following to investigate how to modify bilinear based models to model this uniqueness.

**Definition 2.** *A bilinear model can uniquely model identity means:*

$$\exists! \ \mathbf{R} \in \hat{\mathbb{R}}, \forall \mathbf{h}, \mathbf{t} \in \hat{\mathbb{E}}, \mathbf{h} \neq \mathbf{t}, \ \mathbf{h}^\top \mathbf{R} \mathbf{h} > \mathbf{h}^\top \mathbf{R} \mathbf{t}, \tag{1}$$

*where $\exists!$ is the uniqueness quantification.*

## 4.2   Unit Ball Bilinear Model

From the above examples, it is clear that both embeddings of entities and relations needed to be constrained. Obviously, modeling *identity* requires both entities and relations be restricted, which will reduce expressiveness. To solve this dilemma, we make the cost negligible by minimizing the constraints, one per entity or relation, while modeling the desired property.

To be specific, we normalize the vectors of the entities and the spectral radius of the matrices of the relations to 1 by setting $\hat{\mathbb{E}} = \{\mathbf{e} \mid \|\mathbf{e}\| = 1, \mathbf{e} \in \mathbb{R}^n\}$ and $\hat{\mathbb{R}} = \{\mathbf{R} \mid \rho(\mathbf{R}) = 1, \mathbf{R} \in \mathbb{R}^{n \times n}\}$. We name the proposed model as Unit Ball Bilinear Model (UniBi), since it captures entities in a unit ball as shown in Fig. 1(c). The score function of UniBi is[2]:

$$s(h, r, t) = \mathbf{h}^\top \mathbf{R} \mathbf{t}, \ \|\mathbf{h}\|, \|\mathbf{t}\| = 1, \rho(\mathbf{R}) = 1. \tag{2}$$

We then have the following theorem.

**Theorem 1.** *UniBi is capable to uniquely model identity in terms of definition 2.*

*Proof.* Please refer to the Appendix A.2                                               □

## 4.3   Efficient Modeling

### 4.3.1   Efficient Modeling for Spectral Radius

Although the proposed model has been proven to model *identity* uniquely, it still has a practical disadvantage, since it is difficult to directly represent all matrices whose spectral raidus are 1. In addition, it is also time-consuming to calculate the spectral radius $\rho(\cdot)$ via singular values decomposition (SVD).

To avoid unnecessary decomposition, we divide a relation matrix into three parts $\mathbf{R} = \mathbf{R}_h \mathbf{\Sigma} \mathbf{R}_t$ where $\mathbf{R}_h, \mathbf{R}_t$ are orthogonal matrices and $\mathbf{\Sigma} = \mathrm{Diag}[\sigma_1, \ldots, \sigma_n]$ is a positive semidefinite diagonal matrix. And we maintain the independence of these three components during training. Therefore, it becomes simple to obtain matrices whose spectral radius is 1, that is, $\frac{\mathbf{R}_h \mathbf{\Sigma} \mathbf{R}_t}{\sigma_{max}}$. And we transform the score function Eq. 2 into the following form.

$$s(h, r, t) = \frac{\mathbf{h}^\top \mathbf{R}_h \mathbf{\Sigma} \mathbf{R}_t \mathbf{t}}{\sigma_{max} \|\mathbf{h}\| \|\mathbf{t}\|}, \tag{3}$$

where $\sigma_{max}$ is the maximum among $\sigma_i$.

### 4.3.2   Efficient Modeling for Orthogonal Matrix

In addition, we find that the calculation of the orthogonal matrix is still time-consuming (Tang et al., 2020). To this end, we only consider the diagonal orthogonal block matrix, where each block is a low-dimensional orthogonal matrix. Specifically, we use $k$-dimensional rotation matrices to build $\mathbf{R}_h$ and $\mathbf{R}_t$. Taking $\mathbf{R}_h$ as an example $\mathbf{R}_h = \mathrm{Diag}[\mathbf{SO}(k)_1, \ldots, \mathbf{SO}(k)_{\frac{n}{k}}]$, where $\mathbf{SO}(k)_i$ denotes the $i$-th special orthogonal matrix, that is, the rotation matrix.

The rotation matrix only represents the orthogonal matrices whose determinant are 1 and does not represent the ones whose determinant are $-1$. To this end, we introduce two diagonal sign matrices of $n$-th order $\mathbf{S}_h, \mathbf{S}_t \in \mathbb{S}$ where

$$\mathbb{S} = \{\mathbf{S} \mid \mathbf{S}_{ij} = \begin{cases} \pm 1, & if \ i = j, \\ 0, & if \ i \neq j. \end{cases}\}. \tag{4}$$

---

[2]We present in more detail the similarities and differences between our constraints and those of our predecessors in the Appendix D

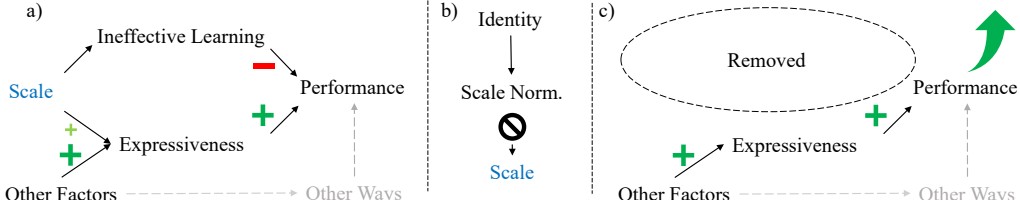

Figure 2: How modeling identity improves the performance. a) An illustration of how scale affects the performance. On the plus side, it has little effect on expressiveness, on the minus side, it causes the ineffective learning. b) Modeling identity requires scale normalization, which removes the effect of scales. c) UniBi improves the performance since it only deals with scales and not the other factors.

Thus, we could rewrite the score function Eq. 3 to

$$s(h, r, t) = \frac{\mathbf{h}^\top \mathbf{R}_h \mathbf{S}_h \mathbf{\Sigma} \mathbf{S}_t \mathbf{R}_t \mathbf{t}}{\sigma_{max} \|\mathbf{h}\| \|\mathbf{t}\|}. \tag{5}$$

However, the sign matrix $\mathbf{S}_h$ and $\mathbf{S}_t$ are discrete. To address this problem, we notice that $\mathbf{S}_h, \mathbf{\Sigma}, \mathbf{S}_t$ can be merged into a matrix $\mathbf{\Xi}$ that

$$\mathbf{\Xi}_{ij} = \begin{cases} s_i s_j \sigma_i, & if\ i = j, \\ 0 & if\ i \neq j. \end{cases} \tag{6}$$

where $s_i = (\mathbf{S}_h)_{ii}$, $s_j = (\mathbf{S}_t)_{jj}$, $i, j = 1, \dots, n$ and $\mathbf{\Xi} = \text{Diag}[\xi_1, \dots, \xi_n]$. Thus, we incorporate the discrete matrices $\mathbf{S}_h, \mathbf{S}_t$ into the continuous matrix $\mathbf{\Xi}$.

$$s(h, r, t) = \frac{\mathbf{h}^\top \mathbf{R}_h \mathbf{\Xi} \mathbf{R}_t \mathbf{t}}{|\xi_{max}| \|\mathbf{h}\| \|\mathbf{t}\|}, \tag{7}$$

where $|\xi_{max}|$ is the maximum among $|\xi_i|$.

### 4.4 OTHER BENEFITS FROM SCALE NORMALIZATION

In addition to theoretical superiority, UniBi is more powerful and interpretable, since modeling identity requires normalizing the scales that barely contain any useful knowledge. On the one hand, it is obvious that modeling identity uniquely needs to avoid the cases in Fig. 1(a) and Fig. 1(b), which requires normalizing the scales of entities and relations. On the other hand, it is counter-intuitive that the scale information is useless for bilinear based models.

Scale information is treated as useless because what really matters is not the absolute values but the relative ranks of scores. And scale contributes nothing to the ranks, since they remain the same after we multiply these scores by a factor greater than zero:

$$s'(h, r, t) = (k_e \mathbf{h})^\top (k_r \mathbf{R})(k_e \mathbf{t}) = k_e^2 k_r (\mathbf{h}^\top \mathbf{R} \mathbf{t}) = k_e^2 k_r \cdot s(h, r, t), \tag{8}$$

where $k_e, k_r > 0$. Therefore, we treat learning on scales as ineffective[3].

#### 4.4.1 PERFORMANCE

As illustrated in Fig. 2, UniBi has better performance, since it prevents ineffective learning with the least constraints. On the one hand, by preventing ineffective learning, UniBi focuses more on learning useful knowledge, which helps improve performance. On the other hand, it pays a negligible cost of expressiveness, since it adds only one equality constraint to each entity or relation, which is ignorable when the dimension $d$ is high.

In other words, although our scale normalization is a double-edged sword, its negative effect is negligible, and thus leads to a better performance. It should be noticed that the loss on expressiveness may outweighs the gain on learning, if scale normalization is replaced by a sticker one. For example, if we constrain the matrix to be orthogonal, the cost of expressiveness is no longer negligible, since an orthogonal matrix requires that each of its singular values be 1, which is $d$ equality constraints.

---

[3]Further discussion in Appendix E.

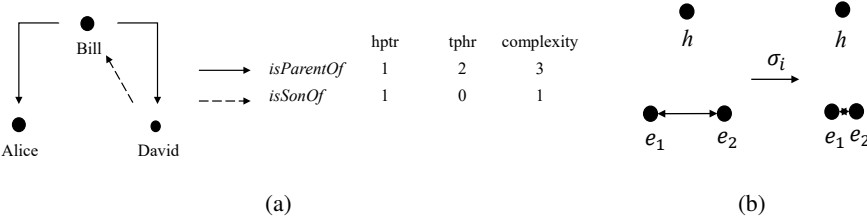

(a)  (b)

Figure 3: Complexity and contraction. (a) A toy example to show how to calculate complexity. (2) the aggregation corresponds to the singular values less than 1.

### 4.4.2 INTERPRETABILITY

In addition to performance, scale normalization also helps us to understand complex relations. Complex relations are defined by whither hptr (*h*ead *p*er *t*ail of a *r*elation) or tphr (*t*ail *p*er *h*ead of a *r*elation) is higher than a special threshold 1.5 (Wang et al., 2014). Therefore, all relations are divided into 4 types, i.e. 1-1, 1-N, N-1, and N-N. However, we think this division is too coarse-grained and suggest a fine-grained continuing metric complexity instead. To better demonstrate this idea, we gave an example in Fig. 3(a) and the definition of complexity as follows.

**Definition 3.** *The complexity of a relation is the sum of its hptr and tphr.*

Intuitively, complex relations are handled by aggregating entities through projection (Wang et al., 2014; Lin et al., 2015), which implies that the higher the complexity of a relation, the stronger its aggregation effect, and vice versa.

We note that this aggregation effect can be well characterized by the relative ratio, or imbalance degree, of singular values of UniBi. For any relation matrix $\mathbf{R} = \mathbf{U}\boldsymbol{\Sigma}\mathbf{V}^\top$, both $\mathbf{U}$ and $\mathbf{V}$ are isometry, and only the singular values of the scaling matrix $\boldsymbol{\Sigma}$ contribute to the aggregation. Moreover, the singular values of UniBi are less than or equal to $1^4$, since the spectral radius, i.e. the maximum singular value, are normalized. This shows a promising correspondence between the singular values of our model and the aggregation and further to the complexity, as demonstrated in Fig. 3(b). Therefore, we could use singular values to represent the complexity of relations, which increases the interpretability of UniBi.

It is worth mentioning that this interpretabity can be transferred to other bilinear based models if they normalize the spectral radius of their relation matrix as UniBi does.

## 5 EXPERIMENT

In this section, we give the experiment settings in Section 5.1. We verify that UniBi is capable to uniquely model *identity*, while previous bilinear based models are not in Section 5.2. UniBi is comparable to previous SOTA bilinear models in the link prediction task, as shown in Section 5.3. In addition, we demonstrate the robustness of UniBi in Section 5.4 and the interpretability about complexity in Section 5.5.

Table 1: Statistics of the benchmark datasets.

| Dataset | $|\mathcal{E}|$ | $|\mathcal{R}|$ | Training | Validation | Test |
|---|---|---|---|---|---|
| WN18RR | 40,943 | 11 | 86,835 | 3,034 | 3,134 |
| FB15k-237 | 14,541 | 237 | 272,115 | 17,535 | 20,466 |
| YAGO3-10-DR | 122,873 | 36 | 732,556 | 3,390 | 3,359 |

---

[4]We discuss this characteristic in more depth from the perspective of group in Appendix F.

## 5.1 Experiment Settings

**Dataset** We evaluate models on three commonly used benchmarks, i.e. WN18RR (Dettmers et al., 2018), FB15k-237 (Toutanova & Chen, 2015) and YAGO3-10-DR (Akrami et al., 2020). They are proposed by removing the reciprocal triples that cause data leakage in WN18, FB15K and YAGO3-10 respectively. Their statistics are listed in Tbl. 1.

**Evaluation metrics** We use Mean Reciprocal Rank (MRR) and Hits@k (k = 1, 3, 10) as the evaluation metrics. MRR is the average inverse rank of the correct entities that are insensitive to outliers. Hits@k denotes the proportion of correct entities ranked above k.

**Baselines** Here we consider two specific versions of UniBi: UniBi-O(2), UniBi-O(3), which use rotation matrices in 2 and 3 dimensions to construct the orthogonal matrix. To be specific, we use the unit complex value and the unit quaternion to model the 2D and 3D rotations using the $2 \times 2$ and $4 \times 4$ matrices, respectively. For more details, see the Appendix G.1.

UniBi is compared with these bilinear models: RESCAL (Nickel et al., 2011), CP (Hitchcock, 1927), ComplEx (Trouillon et al., 2016), and QuatE (Zhang et al., 2019). In addition, it also compared to other models: RotatE (Sun et al., 2019), MurE (Balazevic et al., 2019b) and RotE (Chami et al., 2020), Turcker (Balazevic et al., 2019a) and ConvE (Dettmers et al., 2018).

**Optimization** We adopt the reciprocal setting (Lacroix et al., 2018), which creates a reciprocal relation $r'$ for each $r$ and a new triple $(e_k, r'_j, e_i)$ for each $(e_i, r_j, e_k) \in \mathcal{K}$. Instead of using Cross Entropy directly (Lacroix et al., 2018; Zhang et al., 2020a; 2019), we add an extra scalar $\gamma > 0$ before softmax function. Since UniBi is bounded, it brings an upper bound to loss that makes the model difficult to optimize as discussed by Wang et al. (2017).

$$L = - \sum_{(h,r,t) \in \mathcal{K}_{train}} \log \left( \frac{\exp(\gamma \cdot s(h,r,t))}{\sum_{t' \in \mathcal{E}} \exp(\gamma \cdot s(h,r,t'))} \right) + \lambda \cdot Reg(h,r,t), \qquad (9)$$

where $Reg(h,r,t)$ is the regularization term and $\lambda > 0$ is its factor. Specifically, we only take $Reg(h,r,t)$ as DURA (Zhang et al., 2020a) in experiments, since it significantly outperforms other regularization terms. In addition, $\gamma$ is set to 1 for previous methods and greater than 1 for UniBi. And we set the dimension $n$ to 500. For other details on the implementation, see Appendix G.2.

## 5.2 Modeling Identity Uniquely

In this part, we verify that 1) UniBi is capable to uniquely model *identity* while previous models fail, and 2) both constraints on the embedding of entities and relations are indispensable. We explicitly add *identity* as a new relation to benchmarks and use its corresponding matrix to determine whether the uniqueness is modeled. In particular, this matrix is supposed to converge to the identity matrix $\mathbf{I}$ or a scaled one. To evaluate it, we introduce a new metric imbalance degree $\Delta = (\sum_i \frac{\sigma_i}{\sigma_{max}} - 1)^2$.

We first compare UniBi with CP (Hitchcock, 1927) and RESCAL (Nickel et al., 2011), the least and most expressive bilinear model on FB15k-237. Besides, we also apply DURA (Zhang et al., 2020a) to models to explore whether these methods are able to uniquely model *identity* under extra regularization. As demonstrated in Fig. 4(a), the imbalance degree $\Delta$ of UniBi converges to 0 while others fails, which verifies that UniBi is capable to uniquely model *identity*. In addition, the imbalance of other models decreases to some extent when using DURA, yet they are still unable to uniquely model *identity*. Then, to show that UniBi can uniquely converge to *identity*, we use two matrices $\mathbf{R}_1$ and $\mathbf{R}_2$ to model it independently. As shown in Fig. 4(b), the error between $\mathbf{R}_1$ and $\mathbf{R}_2$ also converges to 0, which means that they all converge to $\mathbf{I}$.

We then perform an ablation study to verify that both the entity constraint (EC) and the relation constraint (RC) are needed to model *identity* uniquely. The experiments show that only using either constraint is not enough to model the uniqueness of *identity*, as illustrated in Fig. 4(c). And this verify the existence of problems shown in Fig. 1(a) and Fig. 1(b).

## 5.3 Main Results

In this part, we demonstrate that the constraints helps UniBi to achieve better performance. We mainly compared our model with previous SOTA models, i.e. CP (Hitchcock, 1927), ComplEx (Trouillon

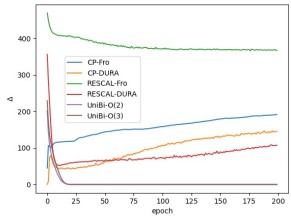
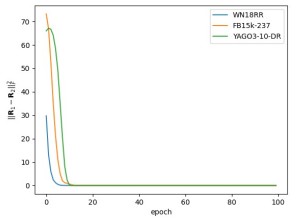
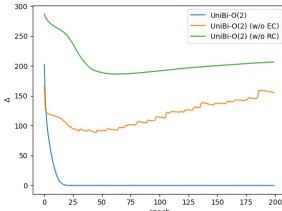

| (a) UniBi V.S. others. | (b) Error between different matrices converges. | (c) Abaltion of constrains. |

Figure 4: UniBi is capable to uniquely model *identity*. (a) the imbalance degree ($\Delta$) of UniBi converges to $0$ while others diverge. (b) The errors between different matrices modeling *identity* converge to $0$ on different datasets. (c) Both entity constrain (EC) and relation constrain (RC) are indispensable for UniBi to model *identity*.

Table 2: Evaluation results on WN18RR, FB15k-237 and YAGO3-10-DR datasets. We reimplement RotE, CP, RESCAL, ComplEx with $n = 500$ and denoted by †, while we take results on WN18RR and FB15k-237 from the origin papers and YAGO3-10-DR from Akrami et al. (2020). Best results are in **bold** while the seconds are underlined.

| Model | WN18RR | | | FB15K-237 | | | YAGO3-10-DR | | |
| | MRR | Hits@1 | Hits@10 | MRR | Hits@1 | Hits@10 | MRR | Hits@1 | Hits@10 |
|---|---|---|---|---|---|---|---|---|---|
| DistMult | 0.43 | 0.39 | 0.49 | 0.241 | 0.155 | 0.419 | 0.192 | 0.133 | 0.307 |
| ConvE | 0.43 | 0.40 | 0.52 | 0.325 | 0.237 | 0.501 | 0.204 | 0.147 | 0.315 |
| TuckER | 0.470 | 0.443 | 0.526 | 0.358 | 0.266 | 0.544 | 0.207 | 0.148 | 0.320 |
| QuatE | 0.488 | 0.438 | 0.582 | 0.348 | 0.248 | 0.550 | - | - | - |
| RotatE | 0.476 | 0.428 | 0.571 | 0.338 | 0.241 | 0.533 | 0.214 | 0.153 | 0.332 |
| MurP | 0.481 | 0.440 | 0.566 | 0.335 | 0.243 | 0.518 | - | - | - |
| RotE | $\underline{0.494}$ | 0.446 | **0.585** | 0.346 | 0.251 | 0.538 | - | - | - |
| CP† | $0.457_{\pm0.001}$ | $0.414_{\pm0.002}$ | $0.549_{\pm0.003}$ | $0.361_{\pm0.001}$ | $0.266_{\pm0.001}$ | $0.551_{\pm0.001}$ | $0.241_{\pm0.001}$ | $0.175_{\pm0.002}$ | $0.370_{\pm0.003}$ |
| ComplEx† | $0.487_{\pm0.002}$ | $0.445_{\pm0.001}$ | $\underline{0.571}_{\pm0.002}$ | $0.363_{\pm0.001}$ | $0.269_{\pm0.001}$ | $0.552_{\pm0.002}$ | $0.238_{\pm0.001}$ | $0.174_{\pm0.002}$ | $0.360_{\pm0.004}$ |
| RESCAL† | $\mathbf{0.495}_{\pm0.001}$ | $\mathbf{0.452}_{\pm0.002}$ | $0.575_{\pm0.002}$ | $0.364_{\pm0.001}$ | $\underline{0.272}_{\pm0.003}$ | $0.547_{\pm0.002}$ | $0.233_{\pm0.003}$ | $0.168_{\pm0.004}$ | $0.360_{\pm0.004}$ |
| UniBi-O(2) | $0.487_{\pm0.002}$ | $0.46_{\pm0.002}$ | $0.566_{\pm0.002}$ | $\mathbf{0.370}_{\pm0.001}$ | $\mathbf{0.274}_{\pm0.001}$ | $\mathbf{0.561}_{\pm0.002}$ | $\mathbf{0.247}_{\pm0.001}$ | $\underline{0.179}_{\pm0.002}$ | $\underline{0.376}_{\pm0.002}$ |
| - w/o constraint | $0.488_{\pm0.001}$ | $\underline{0.447}_{\pm0.002}$ | $0.568_{\pm0.003}$ | $0.361_{\pm0.001}$ | $0.267_{\pm0.001}$ | $0.550_{\pm0.001}$ | $0.242_{\pm0.001}$ | $0.176_{\pm0.001}$ | $0.372_{\pm0.002}$ |
| UniBi-O(3) | $0.492_{\pm0.001}$ | $\mathbf{0.452}_{\pm0.001}$ | $\underline{0.571}_{\pm0.001}$ | $\underline{0.369}_{\pm0.001}$ | $\mathbf{0.274}_{\pm0.001}$ | $\underline{0.561}_{\pm0.001}$ | $\underline{0.246}_{\pm0.001}$ | $\mathbf{0.180}_{\pm0.001}$ | $\mathbf{0.377}_{\pm0.001}$ |
| - w/o constraint | $0.488_{\pm0.001}$ | $0.446_{\pm0.002}$ | $0.567_{\pm0.003}$ | $0.361_{\pm0.001}$ | $0.265_{\pm0.001}$ | $0.550_{\pm0.001}$ | $0.241_{\pm0.001}$ | $0.175_{\pm0.002}$ | $0.370_{\pm0.003}$ |

et al., 2016) and RESCAL (Nickel et al., 2011) using DURA regularization. Although these models have been implemented by Zhang et al. (2020a), the dimensions of CP and ComplEx are very high and have not been tested on YAGO3-10-DR, so we reimplement them in this paper. In addition, we further remove the constraint of UniBi-O(n) as ablations to eliminate the influence of other factors.

In Tbl. 2, UniBi achieves comparable results to previous bilinear based models and the unconstrained versions. UniBi is only slightly and justifiably below RESCAL on WN18RR, since RESCAL needs require much more time and space[5].

## 5.4 UNIBI PREVENTS INEFFECTIVE LEARNING

In this part, we verify the superiority of UniBi comes from preventing ineffective learning. We conduct further comparisons without regularization. In addition, we also adopt EC and RC in Section 5.2 to study the effect of both constraints. All experiments are implemented on WN18RR.

On the one hand, UniBi decreases slightly, while others decrease significantly when the regularization term is removed, as demonstrated in Fig. 5(a). It shows that learning of UniBi is less dependent on extra regularization, since it is better at learning by preventing the ineffective part. On the other hand, we illustrate the MRR metric of UniBi and its ablation models on validation set as epoch grows in Fig. 5(b). It shows that either constrain alleviates overfitting to some extend but fails to prevent the downward sliding behind since the scale of the unconstrained part may diverge. Thus, both constraints are verified to be indispensable for preventing the ineffective learning and performance of UniBi.

---

[5]Detailed in Appendix H.

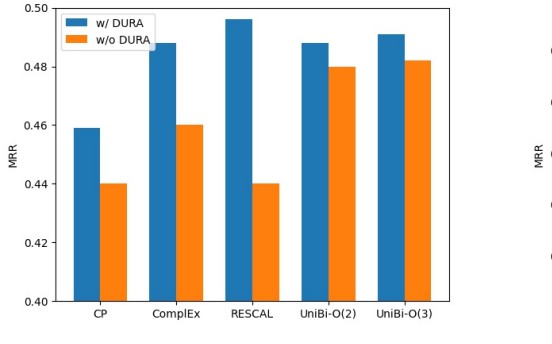

(a) Ablation of regularization.

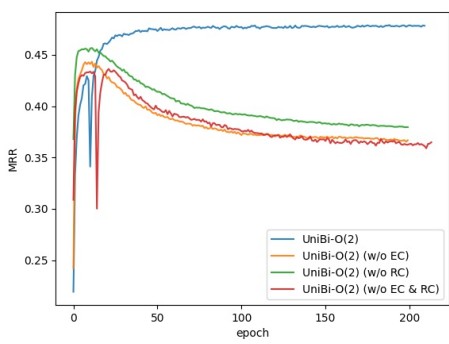

(b) Ablation of constrains.

Figure 5: UniBi benefits from preventing ineffective learning. (a) UniBi maintain its performance without regularization while other models do not. (b) Neither entity constraint (EC) nor relation constrain (RC) alone stops the sliding of performance.

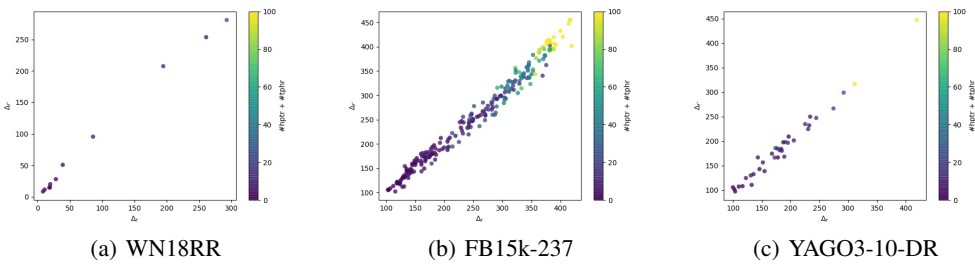

(a) WN18RR        (b) FB15k-237        (c) YAGO3-10-DR

Figure 6: The imbalance degree ($\Delta$) and complexity (# hptr + # tphr) of relations in WN18RR, FB15k-237 and YAGO3-10-DR respectively. Two metric are highly correlated and the imbalance of a relation ($\Delta_r$) and the imbalance of its reciprocal one ($\Delta_{r'}$) are very close.

## 5.5 CORRELATION TO COMPLEXITY

To verify the statement in Section 4.4.2, we study the connection between singular values and the complexity of each relation on three benchmarks, where complexity is calculated following definition 3. Furthermore, we measure the singular values of a relation by the imbalance degree $\Delta$. To differentiate $\Delta$ of a relation $r$ and its reciprocal relation $r'$, we use $\Delta_r$ and $\Delta_{r'}$ to denote them.

As demonstrated in Fig. 6, we find that singular values are highly correlated with the complexity of a relation. Furthermore, we notice that $\Delta_r$ and $\Delta_{r'}$ are very close even if a relation is unbalanced (1-N or N-1), which shows that complexity is handled by aggregation regardless of direction.

## 6 CONCLUSION

In this paper, we reveal that previous bilinear models fail to model the uniqueness of relation *identity* required by general frameworks based on group. To address this problem with negligible cost, we propose UniBi, which has been proven to handle it by capturing entities in a unit ball. Furthermore, UniBi benefits from scale normalization required by modeling the identity uniquely. On the one hand, scale normalization prevents ineffective learning and leads to better performance; on the other hand, it reveals that the relative ratio of singular values corresponds to the complexity of relations and improves interpretability. Therefore, UniBi reconciles the framework and bilinear based models in terms of *identity* and paves the way for further studies of both.

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

## A  PROOFS

### A.1  UNIBI IS BOUNDED

**Proposition 1.** *UniBi is bounded that $s(h, r, t) \in [-1, 1]$.*

*Proof.* By the Cauchy-Schwarz inequality and the fact that the spectral norm is a generalization of the L2 norm, we have the following.

$$\|\mathbf{h}^\top \mathbf{R} \mathbf{t}\| \leq \|\mathbf{h}^\top \mathbf{R}\| \|\mathbf{t}\| \leq \rho(\mathbf{R}) \|\mathbf{h}\| \|\mathbf{t}\| = 1. \tag{10}$$

$\square$

### A.2  PROOF OF THEOREM 1

*Proof.* On the one hand, if $\mathbf{R} = \mathbf{I}$, it is easy to have $\forall \mathbf{h}, \mathbf{t} \in \hat{\mathbb{E}}, \mathbf{h} \neq \mathbf{t}$ that $\mathbf{h}^\top \mathbf{I} \mathbf{h} > \mathbf{h}^\top \mathbf{I} \mathbf{t}$ from the property of cosine.

On the other hand, $\forall \mathbf{R} \in \hat{\mathbb{R}}, \mathbf{R} \neq \mathbf{I}$, we can always give a counterexample. Using singular value decomposition (SVD), we have $\mathbf{R} = \mathbf{U} \boldsymbol{\Sigma} \mathbf{V}^\top$, where $\Sigma = \text{Diag}[\sigma_1, \ldots, \sigma_n]$ with $\sigma_i \geq 0$ and $U, V$ are orthogonal matrices. Since $\rho(\mathbf{R}) = 1$, we have $\sigma_{max} = \max(\sigma_i) = 1$.

Besides, we notice that since $U, V$ are orthogonal matrices that do not change the norm of vectors, we have $\|\mathbf{h}^\top \mathbf{U}\| = \|\mathbf{V}^\top \mathbf{t}\| = 1$ and use $\hat{\mathbf{h}}$ and $\hat{\mathbf{t}}$ to denote $\mathbf{U}^\top \mathbf{h}$ and $\mathbf{V}^\top \mathbf{t}$ for mathematical simplicity. We consider three scenarios and discuss them separately.

(1) If all singular values of $\mathbf{R}$ are equivalent, we have $\boldsymbol{\Sigma} = \mathbf{I}$, and we have:

$$s(h, r, t) = \mathbf{h}^\top \mathbf{U} \boldsymbol{\Sigma} \mathbf{V}^\top \mathbf{t} = \hat{\mathbf{h}}^\top \mathbf{I} \hat{\mathbf{t}} = \hat{\mathbf{h}}^\top \hat{\mathbf{t}}. \tag{11}$$

It is easy to notice that the above equation just goes back to the cosine function, and it has the maximum value when $\hat{\mathbf{h}} = \hat{\mathbf{t}}$ and $\mathbf{U}^\top \mathbf{h} = \mathbf{V}^\top \mathbf{t}$. If $\mathbf{U} \mathbf{V}^\top = \mathbf{I}$, this contradicts the assumption that $\mathbf{R} \neq \mathbf{I}$. If $\mathbf{U} \mathbf{V}^\top \neq \mathbf{I}$, then we have $\mathbf{h} \neq \mathbf{t}$, since $\mathbf{h} = (\mathbf{U}^\top)^{-1} \mathbf{V}^\top \mathbf{t} = \mathbf{U} \mathbf{V}^\top \mathbf{t}$. It means $\mathbf{h}^\top \mathbf{R} \mathbf{h} < \mathbf{h}^\top \mathbf{R} \mathbf{t}$ in this situation.

(2) If not all singular values of $\mathbf{R}$ are equivalent and $\mathbf{U} = \mathbf{V}$, there $\exists i, j \in 1, \ldots, n$ that $\sigma_i \neq \sigma_j$. It may be assumed that $\sigma_i > \sigma_j$. Then we take $\forall \mathbf{h} \in \hat{\mathbb{E}}$ that has $\hat{\mathbf{h}}_j = (\frac{\sigma_i}{\sigma_j} - \epsilon) \hat{\mathbf{h}}_i$ where $\epsilon \in (0, \frac{\sigma_i}{\sigma_j} - 1)$. Then we take

$$\hat{\mathbf{t}}_k = \begin{cases} \hat{\mathbf{h}}_k, & k \neq i, j, \\ \hat{\mathbf{h}}_j, & k = i, \\ \hat{\mathbf{h}}_t, & k = j. \end{cases} \tag{12}$$

It is easy to notice that $\hat{\mathbf{h}} = \mathbf{U}^\top \mathbf{h}, \hat{\mathbf{t}} = \mathbf{U}^\top \mathbf{t}$ and $\mathbf{h} \neq \mathbf{t}$, then we have

$$\begin{aligned} \mathbf{h}^\top \mathbf{R} \mathbf{t} &= \mathbf{h}^\top \mathbf{U} \mathbf{R} \mathbf{U}^\top \mathbf{t} \\ &= \hat{\mathbf{h}}^\top \boldsymbol{\Sigma} \hat{\mathbf{t}} \\ &= \sigma_i \hat{\mathbf{h}}_i \hat{\mathbf{t}}_i + \sigma_j \hat{\mathbf{h}}_j \hat{\mathbf{t}}_j + \sum_{k \neq i, j} \sigma_k \hat{\mathbf{h}}_k \hat{\mathbf{t}}_k \\ &= \sigma_i \hat{\mathbf{h}}_i \hat{\mathbf{h}}_j + \sigma_j \hat{\mathbf{h}}_j \hat{\mathbf{h}}_i + \sum_{k \neq i, j} \sigma_k \hat{\mathbf{h}}_k^2, \end{aligned} \tag{13}$$

similarly, we have

$$
\begin{aligned}
\mathbf{h}^\top \mathbf{R} \mathbf{h} &= \mathbf{h}^\top \mathbf{U} \mathbf{R} \mathbf{U}^\top \mathbf{h} \\
&= \hat{\mathbf{h}}^\top \boldsymbol{\Sigma} \hat{\mathbf{h}} \\
&= \sigma_i \hat{\mathbf{h}}_i^2 + \sigma_j \hat{\mathbf{h}}_j^2 + \sum_{k \neq i,j} \sigma_k \hat{\mathbf{h}}_k^2 .
\end{aligned}
\tag{14}
$$

Then, we use Eq. 14 minus Eq. 13, and we have

$$
\begin{aligned}
& \mathbf{h}^\top \mathbf{R} \mathbf{h} - \mathbf{h}^\top \mathbf{R} \mathbf{t} \\
={}& \left( \sigma_i \hat{\mathbf{h}}_i^2 + \sigma_j \hat{\mathbf{h}}_j^2 + \sum_{k \neq i,j} \sigma_k \hat{\mathbf{h}}_k^2 \right) - \left( \sigma_i \hat{\mathbf{h}}_i \hat{\mathbf{h}}_j + \sigma_j \hat{\mathbf{h}}_j \hat{\mathbf{h}}_i + \sum_{k \neq i,j} \sigma_k \hat{\mathbf{h}}_k^2 \right) \\
={}& \sigma_i (\hat{\mathbf{h}}_i^2 - \hat{\mathbf{h}}_i \hat{\mathbf{h}}_j) + \sigma_j (\hat{\mathbf{h}}_j^2 - \hat{\mathbf{h}}_i \hat{\mathbf{h}}_j) \\
={}& \sigma_i \hat{\mathbf{h}}_i (\hat{\mathbf{h}}_i - \hat{\mathbf{h}}_j) - \sigma_j \hat{\mathbf{h}}_j (\hat{\mathbf{h}}_i - \hat{\mathbf{h}}_j) \\
={}& (\sigma_i \hat{\mathbf{h}}_i - \sigma_j \hat{\mathbf{h}}_j)(\hat{\mathbf{h}}_i - \hat{\mathbf{h}}_j) \\
={}& \left( \sigma_i \hat{\mathbf{h}}_i - \sigma_j (\frac{\sigma_i}{\sigma_j} - \epsilon) \hat{\mathbf{h}}_i \right) \left( \hat{\mathbf{h}}_i - (\frac{\sigma_i}{\sigma_j} - \epsilon) \hat{\mathbf{h}}_i \right) \\
={}& \left( \sigma_i \hat{\mathbf{h}}_i - \sigma_i \hat{\mathbf{h}}_i + \epsilon \sigma_j \hat{\mathbf{h}}_i \right) \left( 1 - \frac{\sigma_i}{\sigma_j} + \epsilon \right) \hat{\mathbf{h}}_i \\
={}& \epsilon \sigma_j \hat{\mathbf{h}}_i^2 (1 - \frac{\sigma_i}{\sigma_j} + \epsilon) \\
<{}& 0,
\end{aligned}
\tag{15}
$$

which means that $\mathbf{h}^\top \mathbf{R} \mathbf{h} < \mathbf{h}^\top \mathbf{R} \mathbf{t}$ in this case.

(3) If not all singular values of $\mathbf{R}$ are equivalent and $\mathbf{U} \neq \mathbf{V}$, there exists $k \in 1, \ldots, n$ such that $\sigma_k = \sigma_{max} = 1$ since $\rho(\mathbf{R}) = 1$. Then we take the following.

$$
\hat{\mathbf{h}}_i = \hat{\mathbf{t}}_i = \begin{cases} 1, & i = k, \\ 0, & i \neq k. \end{cases}
\tag{16}
$$

It should be noted that $\hat{\mathbf{h}} = \hat{\mathbf{t}}$ and $\mathbf{h} \neq \mathbf{t}$ since $\mathbf{U} \neq \mathbf{V}$. Then, we have

$$
\begin{aligned}
\mathbf{h}^\top \mathbf{R} \mathbf{t} &= \mathbf{h}^\top \mathbf{U} \boldsymbol{\Sigma} \mathbf{V}^\top \mathbf{t} \\
&= \hat{\mathbf{h}}^\top \boldsymbol{\Sigma} \hat{\mathbf{t}} \\
&= \sigma_k \\
&= 1.
\end{aligned}
\tag{17}
$$

Since Proposition 1 proves that UniBi is bounded by $[-1, 1]$, we have $\mathbf{h}^\top \mathbf{R} \mathbf{h} \leq 1 = \mathbf{h}^\top \mathbf{R} \mathbf{t}$, which means that $s(h, r, h) > s(h, r, t)$ does not always hold.

In summary, we can conclude that UniBi has $\mathbf{h} \neq \mathbf{t}$, $\mathbf{h}^\top \mathbf{R} \mathbf{h} > \mathbf{h}^\top \mathbf{R} \mathbf{t}$ iff. $\mathbf{R} = \mathbf{I}$, which means that UniBi can model *identity* uniquely in terms of definition 2.

$\square$

### A.3 PROOF OF THEOREM 3

*Proof.* if we have $\|\mathbf{e}\| = 1, \rho(\mathbf{R}) = 1$. The equation for the entity part obviously holds, and by Proposition 1, we have the following.

$$
\|\mathbf{R} \mathbf{e}\| \leq \rho(\mathbf{R}) \|\mathbf{e}\| = \rho(\mathbf{R}) = 1,
\tag{18}
$$

similarly, we have $\|\mathbf{e}^\top \mathbf{R}\| \leq 1$.

Moreover, such a condition will become necessary and sufficient if $\exists \hat{e}$ that either $\|\hat{\mathbf{e}}^\top \mathbf{R}\| = 1$ or $\|\mathbf{R} \hat{\mathbf{e}}\| = 1$.

To prove it, if we have $\|\mathbf{e}^\top \mathbf{R}\| \leq 1$ and $\|\mathbf{Rt}\| \leq 1$, we use SVD to any $\mathbf{R}$ and get $\mathbf{R} = \mathbf{U\Sigma V}$. Then we denote $\sigma = Diag(\sigma)$ where $\sigma$ is the vector of singular values and we have $\forall \mathbf{e}, \|\sigma \cdot \mathbf{e}\| \leq 1$.

If $\exists i, \sigma_i > 1$, we take $i \neq j, e_i = 1, e_j = 0$ to show that $\|\mathbf{e} \cdot \sigma\|$ is larger than 1. Therefore, we have $\forall i, \sigma_i \leq 1$. Moreover, if $\forall i, 0 < \sigma_i < 1$, we take $\bar{\mathbf{e}} = \mathbf{V\hat{e}}$

$$\|\sigma \cdot \bar{\mathbf{e}}\|^2 = \sum_i \sigma_i^2 \bar{e}_i^2 < \sigma_i \bar{e}_i^2 = 1, \tag{19}$$

which also violates $\|\mathbf{Re}\| = \|\mathbf{\Sigma V\hat{e}}\| = \|\sigma \cdot \bar{\mathbf{e}}\| = 1$. Therefore, there $\exists k, \sigma_k = 1$ and we have $\rho(\mathbf{R}) = 1$.

$\square$

### A.4 PROOF OF PROPOSITION 2

*Proof.* If a relation $r$ is invertible, it means there exists an inverse relation $r^{-1}$ that ensure $r \circ r^{-1} = r^{-1} \circ r = identity$, where $\circ$ is the composition of relations. Consider that *identity* relation means that no different entities will appear in a triple under it, or KGs only contain triples like $\forall e \in \mathcal{E}, (e, identity, e)$.

Consider a complex relation that has multiple tail entities for a head entity, that is, $\exists h, t_1, t_2$ $(h, r, t_1), (h, r, t_2)$. In order to remap the entities $t_1$ and $t_2$ back to $h$, the inverse relation $r^{-1}$ has to contain triples that $(t_1, r^{-1}, h), (t_2, r^{-1}, h)$. Although $r \circ r^{-1}$ could map $h$ to $h$, $r^{-1} \circ r$, it fails to map $t_1, t_2$ back to themselves separately since both $(t_1, r^{-1} \circ r, t_1)$ and $(t_1, r^{-1} \circ r, t_2)$ are true. Similarly, if a relation has multiple heads, we can also obtain the above conclusion.

Therefore, all complex relations are inherently non-invertible.

$\square$

## B BACKGROUND OF GROUP AND MONOID

**Definition 4** (Jacobson (2012)). *A monoid is a triple* $(M, p, 1$ *in which $M$ is a non-vacuous set, $p$ is an associative binary composition (or product) in $M$, and 1 is an element of $M$ such that* $p(1, a) = a = p(a, 1)$ *for all $a \in M$*

**Definition 5** (Jacobson (2012)). *A group $G$ ( or (G,p,1)) is a monoid all of whose elements are invertible.*

## C DISCUSS OF DISTANCE BASED MODELS

We mention that distance based model could model *identity* uniquely, and here we give its corresponding proof. Here, we only consider models that can be written in the basic form of $s(h, r, t) = -\|\mathbf{Rh} - \mathbf{t}\|$. Note that translation is also considered in such a form if we take translation as a linear transformation. Furthermore, the model must ensure that $\mathbf{I} \in \hat{\mathbb{R}}$ and $\hat{\mathbb{E}} = \mathbb{R}^n$. Other peculiar scenarios are not considered in this discussion.

**Theorem 2.** *A distance based model $s(h, r, t) = -\|\mathbf{Rh} - \mathbf{t}\|$ with $\mathbf{I} \in \hat{\mathbb{R}}$ and $\hat{\mathbb{E}} = \mathbb{R}^n$ can model identity uniquely.*

*Proof.* On the one hand, if $\mathbf{R} = \mathbf{I}$, it is easy to have

$$-\|\mathbf{Rh} - \mathbf{t}\| = -\|\mathbf{h} - \mathbf{t}\| \leq -\|\mathbf{h} - \mathbf{h}\| = 0, \tag{20}$$

where $\mathbf{h} \neq \mathbf{t}$.

On the other hand, if $\mathbf{R} \neq \mathbf{I}$, if $\mathbf{R}$ is not a singular matrix, we take $\mathbf{t} = \mathbf{Rh}$ and have

$$-\|\mathbf{Rh} - \mathbf{h}\| \leq 0 = -\|\mathbf{Rh} - \mathbf{Rh}\| = -\|\mathbf{Rh} - \mathbf{t}\|. \tag{21}$$

$\square$

Therefore, we prove that distance-based models are born to uniquely model *identity*.

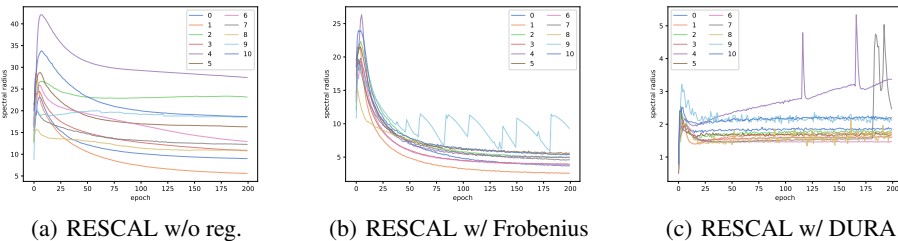

| (a) RESCAL w/o reg. | (b) RESCAL w/ Frobenius | (c) RESCAL w/ DURA |

Figure 7: Why learning on the scales are ineffective. We test RESCAL with 3 setting a) no regularization, b) use Frobenius norm as regularization, c) use DURA as regularization. We use index rather than name to denote different relations for simplicity. And we notice that 1) non-convergence exists in every case, 2) the better the result, the more the scales converge, 3) Regularization cannot stop the fluctuation of scales. (Better view in color, zoom in, note the difference in the vertical coordinates.)

## D   COMPARISON TO PREVIOUS RESTRICTIONS

Although it is true that entity normalization and set $\mathbf{R}$ to orthogonal matrices, which is a special case of $\rho(\mathbf{R}) = 1$, are common in KGE models, our constraint differs from their approach as follows.

(1) The constraint for relation based on the spectral radius, i.e., $\rho(\mathbf{R}) = 1$, is first proposed in our work, which is indispensable and irreplaceable to model the *identity* uniquely without the cost of the performance. If we set $\mathbf{R}$ to be orthogonal while keeping the normalization of entities, the performance will drop significantly, e.g., MRR: $0.488 \rightarrow 0.471$ for UniBi-O(2) on WN18RR.

(2) Only the combination of entity and relation constraints succeeds to model *identity* uniquely, as shown in Figure 4(c). It suggests that constraints on entity and relation should be treated as a whole rather than a combination of two unrelated things.

(3) At first glance, our constraint looks very similar to that of TransR (Lin et al., 2015), but in fact there is a big difference. The constraint of TransR is $\|\mathbf{h}\| \leq 1$, $\|\mathbf{hM}_r\| \leq 1$, and $\|\mathbf{tM}_r\| \leq 1$, and has three differences from ours.

  i) For TransR, $\|\mathbf{hM}_r\| \leq 1$ and $\|\mathbf{tM}_r\| \leq 1$ is the constraint **itself**. In contrast, $\mathbf{R}^\top \mathbf{h} \leq 1$ and $\mathbf{Rt} \leq 1$ is **deduction** of our constraints $\|\mathbf{e}\| = 1$ and $\rho(\mathbf{R}) = 1$.

 ii) TransR does not normalize the entities, and we have shown in Fig. 1(a) that normalization is necessary for a bilinear based method to uniquely model *identity*.

iii) TransR is a distance based model, and as we have shown in Appendix C, distance based models do not need to consider the problem of *identity*, thus the proposal of TransR is difference to us.

Therefore, we believe that our combination of constraints is novel, since it proposes a new constraint for the relation and the two parts are deliberately rather than arbitrarily combined.

## E   INEFFECTIVE LEARNING

Here, we demonstrate that ineffective learning does exist, which means the scale is not only redundant but also harmful. As shown in Fig. 7, we take RESCAL as example to show this phenomenon. And we notice that 1) non-convergence exists in every case, 2) the better the result, the more the scales converge, and 3) regularization cannot stop the fluctuation of scales.

We believe that these cases illustrate, on the one hand, the positive correlation between the constraint scale and the effect, on the other hand, the mere constraint cannot eliminate fluctuations that may interfere with the model learning. Therefore, we think scale is harmful and learning on it is ineffective, and we need a hard constraint rather than regularization term to prevent this completely.

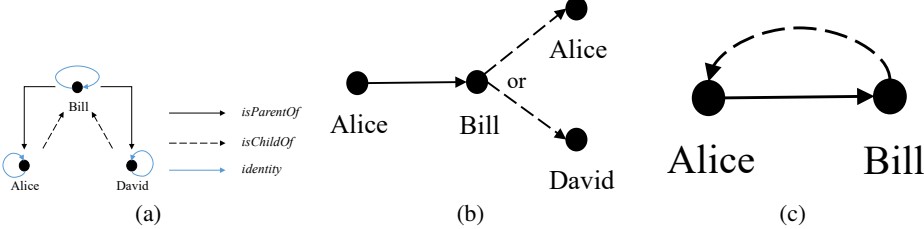

Figure 8: Why some relations are non-invertible. (a) A toy example. (2) True case. (3) False case.

## F FROM GROUP TO MONOID

Since the singular values are less than or equal to 1, some matrices do not have their corresponding inverse elements in $\hat{\mathbb{R}}$. Therefore, UniBi violates the definition of group, which requires that each element has its inverse element inside. However, we argue that some relations are inherently non-invertible and misidentified by previous work (Yang et al., 2020; Sun et al., 2019; Xu & Li, 2019).

For example, in Fig. 8(a), if two relations are inverse elements of each other, their composition must be *identity*. However, we find the combination of *IsParentOf* and *IsChildOf* is not the *identity*, since the child of Alice's parent is "Alice or David" rather than "Alice itself" as demonstrated in Fig. 8(b) and Fig. 8(c).

It should be noted that we are not saying that these two relations are irrelevant, but rather pointing out that their combination does not strictly fit our definition of *identity*. Moreover, the fact that they are both **non-invertible** does not mean that they cannot form the **inversion pattern** (Sun et al., 2019), since definitions of the two concepts are different and do not conflict.

Beyond the specific example, we generalize it to all complex relations that appear in both triples $(e_1, r, e_2)$ and $(e_1, r, e_3)$ (or $(e_3, r, e_2)$), where $e_1, e_2, e_3 \in \mathcal{E}$, and $e_1 \neq e_2 \neq e_3$. Furthermore, we have the following proposition.

**Proposition 2.** $\forall r \in \mathcal{R}$, if $r$ is a complex relation, then $r$ is non-invertible.

*Proof.* Please refer to Appendix A.4. $\qquad\square$

Since these non-invertible relations extensively present in KGs, the requirement that each element has its inverse element in the set should be abandoned. Thus, rather than group, we think monoid is a better structure for relations by their definitions[6].

## G IMPLEMENT DETAILS

### G.1 ROTATION MATRICES OF UNIBI

In Section 5, we propose two variants of UniBi, i.e., UniBi-O(2) and UniBi-O(3). Here, we give the rotation matrix $\mathbf{SO}(k)$ they used specifically.

$$\mathbf{SO}(2) = \begin{bmatrix} x & -y \\ y & x \end{bmatrix}, \ \mathbf{SO}(3) = \begin{bmatrix} p & -q & -u & -v \\ q & p & v & -u \\ u & -v & p & q \\ v & u & -q & p \end{bmatrix}, \tag{22}$$

where $x, y, p, q, u, v \in \mathbb{R}$ and $x^2 + y^2 = 1, p^2 + q^2 + u^2 + v^2 = 1$.

### G.2 HYPERPARAMETERS

We fix the dimension of all models except RESCAL on WN18RR to 500, while RESCAL on WN18RR is set to 256 following Zhang et al. (2020a). We choose Adam (Kingma & Ba, 2015) as

---

[6]Definition 4 and Definition 5.

Table 3: Hyperparameters found by grid search. $\lambda$ is the regularization coefficient, $\gamma$ is the scaling factor, $b$ is the batch size.

| | WN18RR | | | FB15K237 | | | YAGO3-10-DR | | |
| Model | $\lambda$ | $\gamma$ | b | $\lambda$ | $\gamma$ | b | $\lambda$ | $\gamma$ | b |
| --- | --- | --- | --- | --- | --- | --- | --- | --- | --- |
| CP | 1e-1 | 1 | 100 | 5e-2 | 1 | 100 | 5e-3 | 1 | 1000 |
| ComplEx | 1e-1 | 1 | 100 | 5e-2 | 1 | 100 | 1e-2 | 1 | 1000 |
| RESCAL | 1e-1 | 1 | 1000 | 5e-2 | 1 | 1000 | 5e-2 | 1 | 1000 |
| UniBi-O(2) | 2 | 20 | 100 | 2 | 25 | 1000 | 1.5 | 30 | 1000 |
| - w/o constraint | 1e-1 | 1 | 100 | 5e-2 | 1 | 1000 | 5e-2 | 1 | 1000 |
| UniBi-O(3) | 2 | 15 | 100 | 1.5 | 20 | 1000 | 1.5 | 30 | 1000 |
| - w/o constraint | 1e-1 | 1 | 100 | 5e-2 | 1 | 1000 | 5e-2 | 1 | 1000 |

the optimizer and fix the learning rate at $1e - 3$. We set the maximum epochs to 200 and apply the early stopping strategy.

We set the scaling factor $\gamma$ to 1 for all models except UniBi. And we search $\gamma$ from $\{1, 5, 10, 15, 20, 25, 30\}$ for UniBi. For the factor for regularization $\lambda$ we search $\{1, 5e - 1, 1e - 1, 5e - 2, 1e - 2, 5e - 3, 1e - 3\}$ for all models except UniBi and $\{0.5, 1, 1.5, 2, 2.5, 3\}$ for it. Furthermore, we do not search for $\lambda_1$ and $\lambda_2$ in Eq. 25 as in the original paper of DURA (Zhang et al., 2020a). we adopt their settings and set $\lambda_1 = 0.5, \lambda_2 = 1.5$ for ComplEx (Trouillon et al., 2016) and CP (Hitchcock, 1927), while $\lambda_1 = \lambda_2 = 1$ for other cases. The search results are listed on the Tbl. 3. We search for the batch size from $\{100, 1000\}$.

In addition, we implemented all the experiments in PyTorch with a single NVIDIA GeForce RTX 1080Ti graphics card. We repeat each experiment five times and take their means and standard deviations.

For the experiments in Section 5.4, we set $\gamma$ for UniBi without DURA in Section 5.4 to $10, 15, 25$ for WN18RR, FB15k-237, and YAGO3-10-DR for UniBi, respectively.

## H  Time and Space Complexity

Since UniBi needs to calculate some constraints, it spends more space and time. Here, we compare the time and space consumption of UniBi and CP, ComplEx, and RESCAL on FB15k 237, and set the batch size to 1000.

As demonstrated in Fig. 9(a) and Fig. 9(b), UniBi takes a little more time than CP and ComplEx, and more time to compute the regularization since it is more complex. Nevertheless, we noticed that UniBi occupies a space similar to that of CP and ComplEx.

In addition, we note that in Section5.3, UniBi outperforms all bilinear based models except RESCAL on WN18RR. And RESCAL needs significantly more time and space than other models. Therefore, it is not as efficient as other models.

In summary, although UniBi takes a little longer, it is a good balance between complexity and performance.

In future work, we will consider finding other constraints on the relation instead of the spectral radius.

## I  Inherent regularization term

Here we find a necessary condition of UniBi and then deduce its corresponding Lagrangian function.

**Theorem 3.** *UniBi has a necessary condition that $\|\mathbf{e}\| = 1$, $\|\mathbf{Re}\| \leq 1$ and $\|\mathbf{R}^\top \mathbf{e}\| \leq 1$.*

*Proof.* Please refer to the Appendix A.3. □

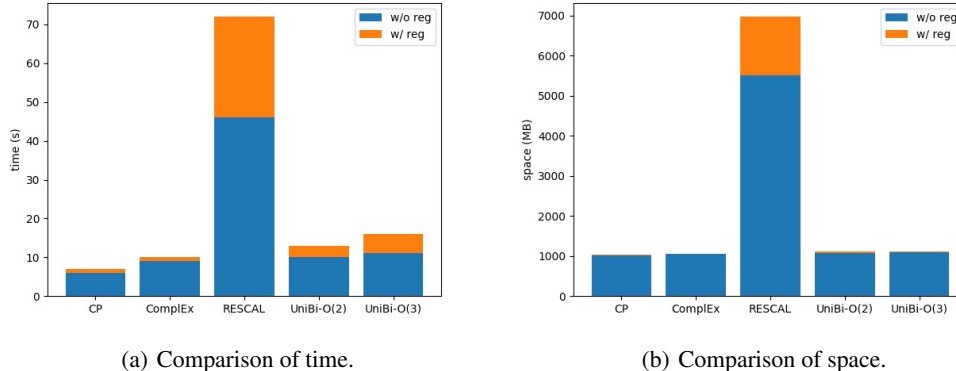

| (a) Comparison of time. | (b) Comparison of space. |

Figure 9: UniBi takes a little more time than other models with linear parameters (CP and ComplEx) and significantly efficient than ones with quadratic parameters (RESCAL).

If we ignore the other implicit constraints, the optimization of UniBi can be rewritten in the form of a constrained optimization problem.

$$
\min_{\mathbf{h},\mathbf{r},\mathbf{t}} \sum_{(h_i,r_j,t_k)\in\mathcal{K}} f(h_i,r_j,t_k) \\
s.t. \|\mathbf{h}_i\|^2 = 1, \|\mathbf{t}_k\|^2 = 1 \\
\|\mathbf{R}_j^\top \mathbf{h}_i\|^2 \le 1, \|\mathbf{R}_j \mathbf{t}_k\|^2 \le 1,
\tag{23}
$$

where $f(h,r,t)$ is a loss function. Furthermore, Eq. 23 corresponds to a Lagrangian function.

$$
\min_{\mathbf{h},\mathbf{r},\mathbf{t},\lambda,\mu} \sum_{(h_i,r_j,t_k)\in\mathcal{K}} f(h_i,r_j,t_k) + \lambda_i^h(\|\mathbf{h}_i\|^2 - 1) + \lambda_k^t(\|\mathbf{t}_k\|^2 - 1) \\
+ \mu_j^h(\|\mathbf{R}_j^\top \mathbf{h}_i\|^2 - 1) + \mu_j^t(\|\mathbf{R}_j \mathbf{t}_k\|^2 - 1) \\
s.t. \lambda_i^h, \lambda_k^t, \mu_j^h, \mu_j^t \ge 0.
\tag{24}
$$

We notice that if we set factors $\lambda_i^h = \lambda_k^t = \lambda\lambda_1$ and $\mu_j^h = \mu_j^t = \lambda\lambda_2$ where $\lambda, \lambda_1, \lambda_2 > 0$, we achieve the following expression from Eq. 24 by discarding constant terms.

$$
\min_{\mathbf{h},\mathbf{r},\mathbf{t}} \sum_{(h_i,r_j,t_k)\in\mathcal{K}} f(h_i,r_j,t_k) + \lambda \left[ \lambda_1(\|\mathbf{h}_i\|^2 + \|\mathbf{t}_k\|^2) + \lambda_2(\|\mathbf{R}_j^\top \mathbf{h}_i\|^2 + \|\mathbf{R}_j \mathbf{t}_k\|^2) \right],
\tag{25}
$$

which is equivalent to the optimization of a unconstrained model under DURA (Zhang et al., 2020a), the best regularization term for bilinear based model before. Therefore, we can get a more general version of DURA (DURA-G for simplicity) from Equ. 24.

$$
\mathbf{DURA\text{-}G} : \lambda_1\|\mathbf{h}_i\|^2 + \lambda_2\|\mathbf{t}_k\|^2 + \lambda_3\|\mathbf{R}_j^\top \mathbf{h}_i\|^2 + \lambda_4\|\mathbf{R}_j \mathbf{t}_k\|^2
\tag{26}
$$

At first glance, this seems to be nothing more than a worthless trick. However, in terms of DURA, DURA-G is nonsense and cannot deduce form its perspective of distance-based models (Please refer to Section 4.3 in the original paper for more details) while making sense in the perspective of Lagrangian function.

Although DURA-G has an additional hyperparameter to search and can lead to better results, we do not use this in experiments for three reasons: 1) for a more fair comparison with previous models, 2) the improvement is marginal, and 3) we prefer not to distract the reader from our key ideas.

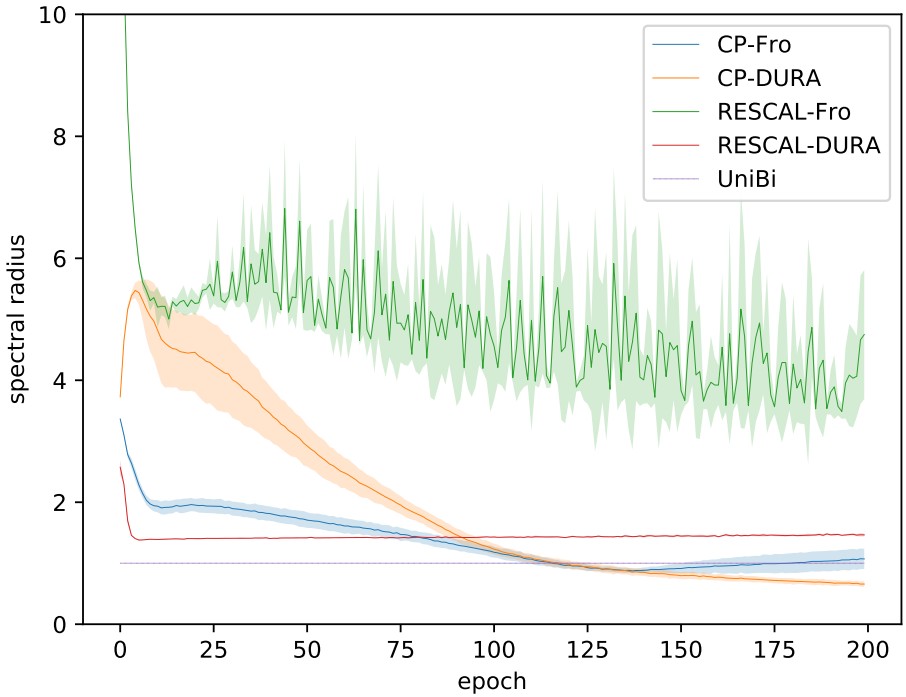

Figure 10: Bilinear based models learning *identity* have different scales. (Better zoom in to see the fluctuation of RESCAL-DURA.)

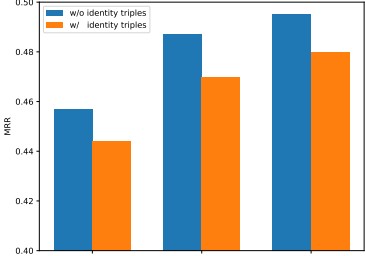

Figure 11: Directly adding identity triples may hurt performance.

Table 4: Statistics of triples of identity and other relations in benchmark datasets.

| Dataset | #Identity | #Others(Avg.) |
|---------|-----------|---------------|
| WN18RR | 40,943 | 7,894 |
| FB15k-237 | 14,514 | 1,148 |
| YAGO3-10-DR | 122,873 | 20,348 |

## J EXTRA EXPERIMENTS OF SCALES

To demonstrate that the phenomenon in Fig. 1(b) happens, we carry out additional experiments by explicitly adding the identity relation explicitly and monitor its scales in different models. We run all experiments five times and take their means and variants. As shown in Fig. 10, we notice that all models except UniBi has fluctuation on the spectral radius of *identity*, which verifies the phenomenon in Fig. 1(b).

## K COMPARISON OF LEARNING AND MODELING IDENTITY

Readers may wonder why not add the identity relation to the training set and take this indirect approach. The reason is that learning on identity per se does not help the performance on other

relations, and may be harmful for the overall results as shown in Fig .11. We think the reason is that the triples of identity are too much compared to the ones of other relations as shown in Tbl., and the model negligence in learning these relations.

