# OpenReview forum: "Rethinking Identity in Knowledge Graph  Embedding"
_ICLR.cc/2023/Conference — Submitted to ICLR 2023_

### Official Review · Reviewer_wvXh · 2022-10-19

**Confidence:** 5
**Correctness:** 3
**Technical Novelty And Significance:** 4
**Empirical Novelty And Significance:** 3
**Recommendation:** 6

**Clarity, Quality, Novelty And Reproducibility:**

**Clarity**
This paper is clearly written and easy to read.

**Quality**
The submission is technically sound. Most of the claims are well supported by experiments or theoretical analyses. Some concerns about the motivation and experiments need to be addressed.

**Novelty**
The proposed method seems novel to me. The authors give a thorough discussion about the difference between their model and related work.

**Reproducibility**
Detailed experimental settings and codes are provided, so the reproducibility looks nice.

**Strength And Weaknesses:**

## Strength
1. The proposed UniBi is simple but effective.
2. UniBi has good theoretical properties and the proof seems to be technically sound.
3. The authors conduct extensive experiments and ablation studies to demonstrate the effectiveness of UniBi.
4. This paper is well-written and easy to follow.
5. Detailed experimental settings and codes are provided, so the reproducibility looks nice.

## Weaknesses
1. **Motivation**
    I think the introduction to why we have to model the identity element can be further improved. Although many existing bilinear models cannot learn a unique identity element, is it necessary for a KGE model?  I have no idea about the relationship between this property and the final performance after reading the submission.
2. **Experiments**
    In Section 5.2, the authors claim that results in Figure 3(c) verify the existence of problems shown in Figures 1(a) and 1(b). However, I am wondering that if this result can verify the problem in Figure 1(b), since the imbalance degree eliminates the impact of the scale of the identity matrix.
3. **Presentation**
    It seems that Definitions 1 and 2 are not used in the main text. Can we move them to the appendix?


**Summary Of The Paper:**

The authors propose a new bilinear knowledge graph embedding model named UniBi by learning a unique identity element in a group. The main idea is to constrain the norm and spectral radius of entity and relation embeddings, respectively. The authors provide thorough analyses to show that UniBi has theoretical superiority, strong robustness and interpretability, and competitive empirical results.

**Summary Of The Review:**

The authors propose an interesting and effective KGE model. The submission is technically sound and easy to read. The experiments are extensive and most of them are convincing. However, I have some concerns about the motivation and some experiments (see Strength And Weaknesses). I am open to raising my score if the authors properly addressed my concerns.

---

> ### Author Response · Authors · 2022-11-18
> **Author Response to Reviewer #4 (wvXh)**
>
> We thank reviewer \#4 for his/her appreciation of our theoretical investigation and valuable concerns. All reference indices are based on the revised version. Our responses are as follows:
>
> > R4-Q1: I have no idea about the relationship between this property and the final performance after reading the submission.
>
> Thanks for your question. This is a very critical and valuable one. We think our responses are insightful and thus add them to the main text in the revision. (We have made many changes, so it's best to check out the revised paper.)
>
> Back to the question. The reason is that our constraint prevents ineffective learning of the model with the least constraints. The former helps to make model learning more focused and effective, and the latter means that we do it at a minimal cost. In the following, we will introduce each of these two points.
>
> **Ineffective learning:** For bilinear-based models, we believe that the scale of either relation or entity contains barely any useful information, at least for link prediction. Therefore, we consider any attempt to optimize scales ineffective. To be more specific, consider a bilinear-based model $s(h,r,t) = \mathbf{h^\top Rt}$. It ranks all candidate entities $e\in\mathcal{E}$ based on their score $s(h,r,e)$, given a query $(h,r,?)$. At this point, we find that scaling the embedding of either relation or entity by any positive number does not change the ranking at all since:
> \begin{align*}
> s'(h,r,t) = (k_e\mathbf{h})^\top (k_r\mathbf{R})(k_e\mathbf{t}) = k_e^2k_r(\mathbf{h^\top Rt}) = k_e^2k_r\cdot s(h,r,t).
> \end{align*}
> This means that learning on the scale is ineffective, and we can concentrate the model more on learning useful knowledge by fixing the scales like our identity constraint, which helps improve performance. And we also show a graph of ineffective learning in Fig. 7.
>
> **Least constrants:** But constraint is not a free lunch, since it helps to learn while also limiting expressiveness. Or in other words, if the constraints are too much, the results may end up with decreases. As an extreme example, we can just simply fix the matrix to $I$ for any relation and completely waste information on the relation types.
>
> From this perspective, our solution is the one with the least constraints because it adds only one equality constraint to each entity and relation. ($\|\mathbf{e}\| = 1$ for entity and $\rho(\mathbf{R}) = 1$ for relation). And we believe that their restrictions on expressiveness are negligible in higher dimensional spaces.
>
> In summary, although our identity constraint is a double-edged sword, its negative effects are negligible and thus lead to better performance.
>
> As an additional note, you may wonder if we can enjoy this benefit by just simply adding the data of identity directly to the training set. Experiments demonstrate that this it does not work and even reduces performance, as shown in Fig. 10. We believe this is due to the fact that the number of triples of identity is much more than the number of other relations and the model thus reduces the attention to them, which are the ones we care. We show the statistics in the following table.
>
> | Dataset  | $\|\mathcal{E}\|$ | $\|\mathcal{R}\|$ | \#Training | \#Identity  | \#Others (avg.) |
> | -- | -- | -- | -- | -- | -- |
> | WN18RR | 40,943 | 11 | 86,835 | 40,943 | 7,894 |
> | FB15k-237 | 14,541 | 237 | 272,115 | 14,514 | 1,148 |
> | YAGO3-10-DR | 122,873 | 36 | 732,556 | 122,873 | 20,348 |
>
> > R4-Q2: I wonder if this result can verify the problem in Figure 1(b) since the imbalance degree eliminates the impact of the scale of the identity matrix.
>
> Thanks for your question. Yes, we can verify this problem, and please check Fig. 10 in the revision. In this figure, we demonstrate that the spectral radius of the matrix of identity fluctuates in every model except UniBi. This means that the learned identity matrix of the same model will have different scales, even the remaining parts being the same.
>
>
> > R4-Q3: Can we move them to the appendix?
>
> Thanks for your suggestion, which increases the compactness of this paper. Therefore, we use a simple example instead and put the formal definitions in Appendix, which is more friendly to readers unfamiliar with the concept, like Reviewer \#2.

---

> > ### Comment · Reviewer_wvXh · 2022-11-25
> > **Thanks for your response**
> >
> > I have read your response and the corresponding part in the revised submission.
> >
> > I agree with you that free scale of entity and relation embeddings may lead to ineffective learning. However, in my opinion, this property is a cause instead of an effect of the ability of modeling identity. In other words, if we eliminate the influence of embedding scale, we may get strong performance and meanwhile model identity better, while the ability of modeling identity itself is unnecessary.
> >
> > *Least constraint* is a strong claim. Is there any evidence to support that it is the least one?
> >
> > It is reasonable that the number of triples of identity is much more than the number of other relations. However, we can drop some of the edges representing identity to make their numbers close.

---

> > > ### Author Response · Authors · 2022-11-26
> > > **Thanks for your question**
> > >
> > > Thanks for your question and our responses are as follows:
> > >
> > > > R4-Q4:  we may get strong performance and meanwhile model identity better, while the ability of modeling identity itself is unnecessary.
> > >
> > > Yes, that is true, but we believe that our main motivation is to model identity theoretically, and the improvement on performance is a secondary object. Further, we believe that even if there is no performance improvement, as long as there is no degradation, it still meets our main motivation.
> > >
> > > In addition, we insist that modeling identity per se is a meaningful motivation. In addition to what we have said in the paper, we think modeling it will give two extra interesting properties. 1) Other potential theoretical benefit: we know that the Euclidean distance and cosine are bijective on a ball, and therefore unification bilinear based models are closer to distance based models, which may lead to a further unification. 2) Potential application in other tasks. As in response to R3 (R3-Q2), we think modeling identity may help bilinear based models in other tasks like entity alignment. Alignment can be treated as finding which entities are identity in different spaces, and
> > > an interesting question is how to measure the identity of entities in cross spaces when the representations of identity are different in each space. In other words, if the representation of identity is unique, then we could use a universal representation of identity to measure the similarity of entities in different spaces. We do not mention this part in the main text because this project has just started.
> > >
> > > > R4-Q5: Least constraint is a strong claim. Is there any evidence to support that it is the least one?
> > >
> > > Yes, from Fig. 1(a) and Fig. 1(b), we all agree that both entity and relation should be restricted, and we add only one equality constraint to each of them. We admit that our method may not be the only solution with $|\mathcal{E}|+|\mathcal{R}|$ equality constraints, but we believe $|\mathcal{E}|+|\mathcal{R}|$ is the least number. To be more specific, if the number of constraints for entities is less than $|\mathcal{E}|$, for example $|\mathcal{E}| - 1$, we could treat an entity as constraint-free while fixing the other entities, which means that the case in Fig. 1(a) is still possible. And for relations we can have a similar result.
> > >
> > > > R4-Q6: we can drop some of the edges representing identity to make their numbers close.
> > >
> > > We tried to reduce the number of identity edges, and find that experiments reinforce the assertion that explicitly adding identity edges will hurt the performance. The experiment is based on RESCAL-DURA on WN18RR, and we only run 1 experiment for each setting in order to quickly response to your question.
> > >
> > > | percent of identity edges | 0% | 10% | 20% | 50% | 100%|
> > > | - | - | - | - | - | - |
> > > | MRR | 0.495 | 0.492 | 0.493 | 0.489 | 0.480|
> > >
> > >
> > >
> > > And we believe the results prove that the improvement is not by implicitly using more training data (identity edges).

---

> ### Author Response · Authors · 2022-11-24
> **Further discussion with Reviewer wvXh (denoted as R4)**
>
> Dear reviewer wvXh
>
> We thank you for the precious review time and valuable comments. We have provided corresponding responses and results, which we believe have covered your concerns. We hope to further discuss with you whether or not your concerns have been addressed. Please let us know if you still have any unclear parts of our work.
>
> Best,
>
> Paper 3461 Authors.

---

### Official Review · Reviewer_Nd46 · 2022-10-24

**Confidence:** 3
**Correctness:** 3
**Technical Novelty And Significance:** 3
**Empirical Novelty And Significance:** 3
**Recommendation:** 6

**Clarity, Quality, Novelty And Reproducibility:**

- Clarity: All notations are gradually introduced and illustrated. Didactically the paper is well-written.

- Novelty: The proposed approach is exploring a new type of models and is novel.

- Reproducibility: Attached to the submission is code with implementation. If shared publicly it will ensure the reproducibility.

**Strength And Weaknesses:**

Strengths:

- The approach is theoretically grounded.

- Experiments are conducted in comparison with multiple strong baselines and show that the method outperforms or perform on par with them.

Weaknesses:

- The improvements are relatively small in absolute values.

- No discussion of applications of the proposed method to other tasks.

**Summary Of The Paper:**

The paper proposes a new approach to learning of graph embeddings. Namely, the theoretical shortcoming of the common bilinear models related to non-uniqueness of identity is addressed. A new theoretically well-grounded model is proposed. Experiments suggest viability fo the described approach.

**Summary Of The Review:**

The main drawback of the paper is that it addresses somewhat 'over researched' topic of link prediction. Yet it uses interesting technically and theoretically grounded approach. Maybe it can be generalised and applied to other less common and innovative task?

---

> ### Author Response · Authors · 2022-11-18
> **Author Response to Reviewer #3 (Nd46)**
>
> We thank reviewer \#3 for his/her helpful suggestions and valuable concerns. All reference indices are based on the revised version. Our responses are as follows:
>
> > R3-Q1: The improvements are relatively small in absolute values.
>
> Thanks for your problem. The slight improvement can be explained from two perspectives. On the one hand, the performances on these datasets are reaching a glass ceiling. To demonstrate this phenomenon, we added some results in recent years for comparison (we did not compare them to those in the main text because this paper is mainly dealing with bilinear-based models).
>
> | Model | WN18RR | FB15k-237 |
> | - | - | - |
> | **Bilinear Based** |
> | ComplEx-DURA (2020) [c1] | 0.487 | 0.363 |
> | RESCAL-DURA (2020) [c1] | 0.495 | 0.364 |
> | UniBi-3 (Ours) | 0.492 | 0.369 |
> | **Distance Based** |
> | RotE (2020) [c2] | 0.494 | 0.346 |
> | HAKE (2020) [c3] | 0.497 | 0.346 |
> | GIE (2022) [c4] | 0.491 | 0.362 |
> | **Neural Network Based** |
> | DisenKGAT (2021) [c5] | 0.486 | 0.368 |
> | HittER (2021) [c6] | 0.485 | 0.373 |
> | Relphormer (2022) [c7] | 0.495 | 0.371 |
>
> On the other hand, we focus on modeling the identity property theoretically rather than achieving much improvement practically. And if we wanted to achieve higher performance, we would use the generalized DURA [c1], which has more hyperparameters to search.
>
> To show the difference between DURA and its generalization (DURA-G for simplicity) that we do not mention explicitly, we notice that DURA is actually the latter part of Equ.(25):
> \begin{equation}
>     \lambda\left[\lambda_1(\|\mathbf{h}_i\|^2 + \|\mathbf{t}_k\|^2)
>     + \lambda_2(\|\mathbf{R}_j^\top \mathbf{h}_i\|^2 + \|\mathbf{R}_j
>     \mathbf{t}_k\|^2)\right]
> \end{equation}
> As we have mentioned, this is achieved by setting factors $\lambda_i^h = \lambda_k^t = \lambda\lambda_1$ and $\mu_j^h = \mu_j^t = \lambda\lambda_2$ in Equ.(24). Similarly, we take the loosened later part in this equation:
> \begin{equation}
>   \lambda_i^h(\|\mathbf{h}_i\|^2-1) +\lambda^t_k(\|\mathbf{t}_k\|^2 - 1)
>     + \mu^h_j(\|\mathbf{R}_j^\top \mathbf{h}_i\|^2 - 1) + \mu^t_j(\|\mathbf{R}_j\mathbf{t}_k\|^2 - 1).
> \end{equation}
> Then we get DURA-G by treating $\lambda$ and $\mu$ as hyperparameters and dropping the constants.
> \begin{equation}
>   \lambda_i^h(\|\mathbf{h}_i\|^2) +\lambda^t_k(\|\mathbf{t}_k\|^2)
>     + \mu^h_j(\|\mathbf{R}_j^\top \mathbf{h}_i\|^2) + \mu^t_j(\|\mathbf{R}_j\mathbf{t}_k\|^2).
> \end{equation}
>
> At first glance, this seems to be nothing more than a worthless trick. However, in terms of DURA, DURA-G is nonsense and cannot deduce from its perspective of distance-based models (Please refer to Section 4.3 in the original paper for more details) while making sense in the perspective of the Lagrangian function.
>
> > R3-Q2: No discussion of applications of the proposed method to other tasks.
>
> Thanks for your further interest. We considered this possibility but left it out of the paper because it was still under consideration. And we are considering two future directions: 1) Although a bilinear-based model per se is not common in the era of deep learning, many models contain bilinear calculations, such as GNN [c8]. 2) Identity may be an interesting property for another task in the knowledge graph, such as knowledge graph alignment.
>
> **Reference**
>
> c1: Zhang Z, Cai J, Wang J. Duality-induced regularizer for tensor factorization based knowledge graph completion[J]. Advances in Neural Information Processing Systems, 2020, 33: 21604-21615.
>
> c2: Chami I, Wolf A, Juan D C, et al. Low-Dimensional Hyperbolic Knowledge Graph Embeddings[C]//Proceedings of the 58th Annual Meeting of the Association for Computational Linguistics. 2020: 6901-6914.
>
> c3: Zhang Z, Cai J, Zhang Y, et al. Learning hierarchy-aware knowledge graph embeddings for link prediction[C]//Proceedings of the AAAI Conference on Artificial Intelligence. 2020, 34(03): 3065-3072.
>
> c4: Cao Z, Xu Q, Yang Z, et al. Geometry Interaction Knowledge Graph Embeddings[C]//AAAI Conference on Artificial Intelligence. 2022.
>
> c5: Wu J, Shi W, Cao X, et al. DisenKGAT: knowledge graph embedding with disentangled graph attention network[C]//Proceedings of the 30th ACM International Conference on Information \& Knowledge Management. 2021: 2140-2149.
>
> c6: Chen S, Liu X, Gao J, et al. HittER: Hierarchical Transformers for Knowledge Graph Embeddings[C]//Proceedings of the 2021 Conference on Empirical Methods in Natural Language Processing. 2021: 10395-10407.
>
> c7: Bi Z, Cheng S, Zhang N, et al. Relphormer: Relational Graph Transformer for Knowledge Graph Representation[J]. arXiv preprint arXiv:2205.10852, 2022.
>
> c8: Zhu H, Feng F, He X, et al. Bilinear graph neural network with neighbor interactions[C]//Proceedings of the Twenty-Ninth International Conference on International Joint Conferences on Artificial Intelligence. 2021: 1452-1458.

---

> ### Author Response · Authors · 2022-11-24
> **Further discussion with Reviewer Nd46 (denoted as R3)**
>
> Dear reviewer Nd46
>
> We thank you for the precious review time and valuable comments. We have provided corresponding responses and results, which we believe have covered your concerns. We hope to further discuss with you whether or not your concerns have been addressed. Please let us know if you still have any unclear parts of our work.
>
> Best,
>
> Paper 3461 Authors.

---

### Official Review · Reviewer_52Mc · 2022-10-25

**Confidence:** 3
**Clarity, Quality, Novelty And Reproducibility:** The introduction and the abstract can…
**Correctness:** 3
**Technical Novelty And Significance:** 3
**Empirical Novelty And Significance:** 3
**Recommendation:** 5

**Strength And Weaknesses:**

Strength:

1.This paper focuses on an interesting problem of KGE and finds a new problem for bilinear-based models.

2.Experimental results show that the proposed method can obtain better performance. This work also provides ablation studies with analysis.

3.The proposed approach also provides a theoretical analysis that demonstrates the superiority.

Weaknesses:

1.The introduction and the abstract can be carefully revised, which is hard to follow. Lots of background is missing, which makes the readers very confused.

2.The experimental results show that the improvement is very small; why? And can you explain it? Could you provide some case studies？

3.Some small typos such as  ", Several works also" >, "several works also"

**Summary Of The Paper:**

This paper targets the special element identity that uniquely corresponds to the relation identity in KGs and finds that bilinear-based models cannot model this uniqueness. This paper studies the required conditions and proposes a solution named Unit Ball Bilinear Model (UniBi). In addition to its theoretical superiority, UniBi is more robust and interpretable. Experiments demonstrate that UniBi models the uniqueness without the cost of performance and verify its robustness and interpretability.

**Summary Of The Review:**

Overall, I think this paper do have some merit, but the writing can be improved and the more experiments  can make this paper stronger.

---

> ### Author Response · Authors · 2022-11-18
> **Author Response to Reviewer #2 (52Mc) (Part 1/2)**
>
> We thank reviewer \#2 for his/her appreciation of our paper and valuable concerns. All reference indices are based on the revised version. our responses are as follows:
>
> > R2-Q1: Lots of background is missing, which makes the readers very confused.
>
> Thanks for your suggestion. We think the confusing part is the idea of group. Therefore, we add an informal definition and a simple example in the introduction and background.
>
> > R2-Q2: The experimental results show that the improvement is very small; why? And can you explain it? Could you provide
> some case studies?
>
> Thanks for your question. We answer the two questions respectively.
>
> **Explanation of improvement:**
> The slight improvement can be explained from two perspectives. On the one hand, the performances on these datasets are reaching a glass ceiling. To demonstrate this phenomenon, we added some results in recent years for comparison (we did not compare them to those in the main text because this paper is mainly dealing with bilinear-based models).
>
> | Model | WN18RR | FB15k-237 |
> | - | - | - |
> | **Bilinear Based** |
> | ComplEx-DURA (2020) [c1] | 0.487 | 0.363 |
> | RESCAL-DURA (2020) [c1] | 0.495 | 0.364 |
> | UniBi-3 (Ours) | 0.492 | 0.369 |
> | **Distance Based** |
> | RotE (2020) [c2] | 0.494 | 0.346 |
> | HAKE (2020) [c3] | 0.497 | 0.346 |
> | GIE (2022) [c4] | 0.491 | 0.362 |
> | **Neural Network Based** |
> | DisenKGAT (2021) [c5] | 0.486 | 0.368 |
> | HittER (2021) [c6] | 0.485 | 0.373 |
> | Relphormer (2022) [c7] | 0.495 | 0.371 |
>
> On the other hand, we focus on modeling the identity property theoretically rather than achieving much improvement practically. And if we wanted to achieve higher performance, we would use the generalized DURA [c1], which has more hyperparameters to search.
>
> To show the difference between DURA and its generalization (DURA-G for simplicity) that we do not mention explicitly, we notice that DURA is actually the latter part of Equ.(25):
> \begin{equation}
>     \lambda\left[\lambda_1(\|\mathbf{h}_i\|^2 + \|\mathbf{t}_k\|^2)
>     + \lambda_2(\|\mathbf{R}_j^\top \mathbf{h}_i\|^2 + \|\mathbf{R}_j
>     \mathbf{t}_k\|^2)\right]
> \end{equation}
> As we have mentioned, this is achieved by setting factors $\lambda_i^h = \lambda_k^t = \lambda\lambda_1$ and $\mu_j^h = \mu_j^t = \lambda\lambda_2$ in Equ.(24). Similarly, we take the loosened later part in this equation:
> \begin{equation}
>   \lambda_i^h(\|\mathbf{h}_i\|^2-1) +\lambda^t_k(\|\mathbf{t}_k\|^2 - 1)
>     + \mu^h_j(\|\mathbf{R}_j^\top \mathbf{h}_i\|^2 - 1) + \mu^t_j(\|\mathbf{R}_j\mathbf{t}_k\|^2 - 1).
> \end{equation}
> Then we get DURA-G by treating $\lambda$ and $\mu$ as hyperparameters and dropping the constants.
> \begin{equation}
>   \lambda_i^h(\|\mathbf{h}_i\|^2) +\lambda^t_k(\|\mathbf{t}_k\|^2)
>     + \mu^h_j(\|\mathbf{R}_j^\top \mathbf{h}_i\|^2) + \mu^t_j(\|\mathbf{R}_j\mathbf{t}_k\|^2).
> \end{equation}
>
> At first glance, this seems to be nothing more than a worthless trick. However, in terms of DURA, DURA-G is nonsense and cannot deduce from its perspective of distance-based models (Please refer to Section 4.3 in the original paper for more details) while making sense in the perspective of the Lagrangian function.
>
> **Curves rather than case study:**  Since UniBi is not gaining from handling an explicit problem like complex relations [c8, c9] or some patterns [c10, c11]. we prefer curves rather than cases to explain the improvements.
>
> As we have shown in Section 4.4.1 in the revision, the improvement comes from preventing ineffective learning on scales. (This is a new idea in revision, and we prefer to read it in the paper since we add some graphs for understanding).
>
> Although it is hard to illustrate how preventing ineffective learning, we demonstrate why ineffective learning hurts performance in Fig. 7.
>
> > R2-Q3: Some small typos such as ", Several works also" >, "several works also"
>
> Thanks for your reminder. We have made the appropriate changes in the paper.

---

> ### Author Response · Authors · 2022-11-18
> **Author Response to Reviewer #2 (52Mc) (Part 2/2)**
>
> **Reference**
>
> c1: Zhang Z, Cai J, Wang J. Duality-induced regularizer for tensor factorization based knowledge graph completion[J]. Advances in Neural Information Processing Systems, 2020, 33: 21604-21615.
>
> c2: Chami I, Wolf A, Juan D C, et al. Low-Dimensional Hyperbolic Knowledge Graph Embeddings[C]//Proceedings of the 58th Annual Meeting of the Association for Computational Linguistics. 2020: 6901-6914.
>
> c3: Zhang Z, Cai J, Zhang Y, et al. Learning hierarchy-aware knowledge graph embeddings for link prediction[C]//Proceedings of the AAAI Conference on Artificial Intelligence. 2020, 34(03): 3065-3072.
>
> c4: Cao Z, Xu Q, Yang Z, et al. Geometry Interaction Knowledge Graph Embeddings[C]//AAAI Conference on Artificial Intelligence. 2022.
>
> c5: Wu J, Shi W, Cao X, et al. DisenKGAT: knowledge graph embedding with disentangled graph attention network[C]//Proceedings of the 30th ACM International Conference on Information \& Knowledge Management. 2021: 2140-2149.
>
> c6: Chen S, Liu X, Gao J, et al. HittER: Hierarchical Transformers for Knowledge Graph Embeddings[C]//Proceedings of the 2021 Conference on Empirical Methods in Natural Language Processing. 2021: 10395-10407.
>
> c7: Bi Z, Cheng S, Zhang N, et al. Relphormer: Relational Graph Transformer for Knowledge Graph Representation[J]. arXiv preprint arXiv:2205.10852, 2022.
>
>
> c8: Lin Y, Liu Z, Sun M, et al. Learning entity and relation embeddings for knowledge graph completion[C]//Twenty-ninth AAAI conference on artificial intelligence. 2015.
>
> c9: Wang Z, Zhang J, Feng J, et al. Knowledge graph embedding by translating on hyperplanes[C]//Proceedings of the AAAI conference on artificial intelligence. 2014, 28(1).
>
> c10: Liu H, Wu Y, Yang Y. Analogical inference for multi-relational embeddings[C]//International conference on machine learning. PMLR, 2017: 2168-2178.
>
> c11: Sun Z, Deng Z H, Nie J Y, et al. Rotate: Knowledge graph embedding by relational rotation in complex space[J]. arXiv preprint arXiv:1902.10197, 2019.

---

> ### Author Response · Authors · 2022-11-24
> **Further discussion with Reviewer 52Mc (denoted as R2)**
>
> Dear reviewer 52Mc
>
> We thank you for the precious review time and valuable comments. We have provided corresponding responses and results, which we believe have covered your concerns. We hope to further discuss with you whether or not your concerns have been addressed. Please let us know if you still have any unclear parts of our work.
>
> Best,
>
> Paper 3461 Authors.

---

### Official Review · Reviewer_VDJN · 2022-10-26

**Confidence:** 4
**Correctness:** 2
**Technical Novelty And Significance:** 3
**Empirical Novelty And Significance:** 2
**Recommendation:** 5

**Clarity, Quality, Novelty And Reproducibility:**

Most part of the paper is clear. What is not very clear is why the proposed method can lead to experimental improvements.
The paper proposes a special schema of normalization to satisfy the identity properties. Considering the fact that embedding normalization has been often used in the literature, the novelty of the paper is limited (unless the difference is shown to be critical, which is not done in the paper).
It may be relatively difficult to reproduce the method, as the source code is not provided.

After rebuttal:
It is true that the code is mentioned in appendix. Reproducibility is good.

**Strength And Weaknesses:**

The analysis of the identity properties is interesting. The proposed solution is simple, but makes sense.
From a theoretical point of view, this investigation is interesting.
From the perspective of experiments, the link between identity and better performance is not made. Despite the ablation studies, it is still difficult to see why satisfying identity would lead to better relation prediction performance. In other words, why the identity constraint would make the embeddings better representations of the entities and relations? It would be necessary to establish this connection for (1) a better understanding of the experimental results; (2) a better motivation for the proposed approach.
As stated in the paper (Appendix), normalization has been applied in some previous work. The only difference is that this paper uses a combination of entity normalization and spectral radius normalization for relations. In the demonstration that the existing methods do not satisfy identity, entities are not normalized. Does the demonstration still hold when entities are normalized?
The citations are in a form difficult to read. Instead of "QuatE Zhang et al. (2019)", it is better to write "QuatE (Zhang et al. 2019)".
The statements in the paper tend to be formal, which may make it more difficult to understand.

After author's rebuttal:
- In the rebuttal, the authors have explained some of the questions above. In particular, the motivation is better articulated.
The authors also mentioned the test about the impact of using the proposed normalization instead of another one. One experimental result is mentioned in appendix D, saying that MRR is reduced from 0.488 to 0.471 on WN18RR if spectral radius is replaced by orthogonal one. I think this should be one of the main results of the paper - showing the practical impact when the proposed normalization is replaced by another one. I would suggest to extend this experimental result to all the datasets, and incorporate them into the main body of the paper.
- Despite the theoretical advantages, the overall effectiveness reported in Table 2 only shows marginal improvements. In particular, one can see that an existing approach RESCAL can obtain 0.495 in MRR on WN18RR. By comparing this with UniBi, it does not seem that the proper normalization, as proposed in this paper, is the most important (or the only key) factor that affect the effectiveness.
In summary, while the motivation is better explained, the experimental evidence about the importance of the proposed normalization is still weak.

**Summary Of The Paper:**

This paper examines the properties of identity in bilinear models for knowledge encoding. To satisfy the properties, a modified bilinear model is proposed, in which the entity and relation embeddings are normalized. This form a unit ball. It is shown that the desired properties of unity can be met.
In the experiments on 3 datasets for relation prediction, the proposed model outperforms slightly the existing bilinear models and other knowledge graph embedding models.


**Summary Of The Review:**

The paper examines a specific property - identity. While theoretically interesting, its practical implication is not well explained. The experimental results are thus difficult to interpret. Compared to the literature (which also uses normalization), the novelty is limited, and the importance of the difference with the existing normalization approaches is unclear.

---

> ### Author Response · Authors · 2022-11-18
> **Author Response to Reviewer \#1 (VDJN)  (Part 1/2)**
>
> We thank reviewer \#1 for his/her appreciation of our theoretical investigation and the valuable questions. We think some of them are very constructive and have revised the paper accordingly. All reference indices are based on the revised version. Our responses are as follows:
>
> > R1-Q1: The source code is not provided.
>
> Thank you for your interest in our source code, and we included it in the supplementary material when we submitted the paper. In addition, both reviewers \#3 and \#4 have mentioned that the reproducibility looks nice. We first answer this question to strengthen the credibility of the experiments we have presented in the paper.
>
> > R1-Q2: why the identity constraint would make the embeddings better representations of the entities and relations?
>
> Thanks for your question. This is a very critical and valuable one. We think our responses are insightful and thus add them to the main text in the revision. (We have made many changes, so it's best to check out the revised paper.)
>
> Back to the question. The reason is that our constraint prevents ineffective learning of the model with the least constraints. The former helps to make model learning more focused and effective, and the latter means that we do it at a minimal cost. In the following, we will introduce each of these two points.
>
> **Ineffective Learning: **
> For bilinear-based models, we believe that the scale of either relation or entity contains barely any useful information, at least for completion. Therefore, we consider any attempt to optimize scales as ineffective. To be more specific, consider a bilinear based model $s(h,r,t) = \mathbf{h^\top Rt}$. It ranks all candidate entities $e\in\mathcal{E}$ based on their score $s(h,r,e)$, given a query $(h,r,?)$. At this point, we find that scaling the embedding of either relation or entity by any positive factor does not change the ranking at all, since:
> \begin{align*}
> s'(h,r,t) = (k_e\mathbf{h})^\top (k_r\mathbf{R})(k_e\mathbf{t}) = k_e^2k_r(\mathbf{h^\top Rt}) = k_e^2k_r\cdot s(h,r,t).
> \end{align*}
> This means that learning on the scale is ineffective, and we can concentrate the model more on learning useful knowledge by fixing the scales like our identity constraint, which helps improve performance. And we also demonstrate ineffective learning in Fig. 7.
>
> **Least Constraints: **
>  But constraint is not a free lunch, since it helps to learn while also limiting expressiveness. Or in other words, if the constraints are too much, the results may end up with decreases. As an extreme example, we can just simply fix the matrix to $I$ for any relation and completely waste information on the relation types.
>
> From this perspective, our solution is the one with the least constraints because it adds only one equality constraint to each entity and relation. ($\|\mathbf{e}\| = 1$ for entity and $\rho(\mathbf{R}) = 1$ for relation). We believe that such restriction on expressiveness is negligible in higher-dimensional spaces.
>
> In summary, although our identity constraint is a double-edged sword, its negative effects are negligible and thus lead to better performance. And we also demonstrate this idea in Fig. 2.
>
> As an additional note, you may wonder if we can enjoy this benefit by simply adding the data of identity directly to the training set. Experiments demonstrate that this does not work and even reduces performance, as shown in Fig. 10. We believe this is due to the fact that the number of triples of identity is much more than the number of other relations and the model thus reduces the attention to them, which are the ones we really care. We show the statistics in the following table.
>
> | Dataset  | $\|\mathcal{E}\|$ | $\|\mathcal{R}\|$ | \#Training | \#Identity  | \#Others (avg.) |
> | -- | -- | -- | -- | -- | -- |
> | WN18RR | 40,943 | 11 | 86,835 | 40,943 | 7,894 |
> | FB15k-237 | 14,541 | 237 | 272,115 | 14,514 | 1,148 |
> | YAGO3-10-DR | 122,873 | 36 | 732,556 | 122,873 | 20,348 |

---

> ### Author Response · Authors · 2022-11-18
> **Author Response to Reviewer \#1 (VDJN) (Part 2/2)**
>
> > R1-Q3: The only difference is that this paper uses a combination of entity normalization and spectral radius normalization for
> relations.
>
> Thanks for your question. Here we want to emphasize two things.
>
> The first is that our combination is not trivial. Beyond what we have shown in Appendix D, we further show differences in the perspective of ineffective learning as we discussed in R1-Q2. If we only constrain either relation or entity, ineffective learning will still exist since the scale of the other is not restricted. And the model may be caught in a vicious competition of scales, as we have shown in Fig. 7, especially when the regularization is absent or weak. Besides, We think this also explains the rollback in the ablation Fig. 5(b).
>
> The second is the novelty of the spectral norm in KGE. As stated in the Appendix, normalizing the spectral radius is new and contains orthogonalization commonly used in previous works [c1, c2]. And the difference between spectral norm and orthogonalization is not trivial. On the one hand, spectral norm assigns only one equality constraint to each relation; On the other hand, orthogonalization means $d$ constraints since it requires each singular value to be 1. Therefore, the impact of orthogonalization on expressiveness is not negligible, as the performance drops from 0.488 to 0.471 shown in Appendix D.
>
> > R1-Q4: Does the demonstration still hold when entities are normalized?
>
> Thanks for your question. Although some methods using orthogonal matrices can model such property with entity normalization, like QuatE, their performances are lower than the ones using spectral norm, as we have shown in Appendix D and explained in R1-Q2 and R1-Q3.
>
> Moreover, we think this question is, in fact, an extension of R1-Q3, asking what the difference is between "orthogonal + entity norm" and "spectral norm + entity norm". We have to admit that, in terms of modeling identity, the former is not elegant but feasible since it is a special case of the latter. Therefore, we modified the motivation as you suggested to emphasize the differences in terms of expressiveness and performance and the special features of our approach. Thanks again for your question.
>
> > R1-Q5: The citations are in a form difficult to read.
>
> Thanks for your suggestion, and we have changed the corresponding citations. We think such a suggestion is helpful to improve the clarity of our paper.
>
> **Reference**
>
> c1: Zhiqing Sun, Zhi-Hong Deng, Jian-Yun Nie, and Jian Tang. Rotate: Knowledge graph embedding by relational rotation in complex space. In ICLR, 2019.
>
> c2: Hanxiao Liu, Yuexin Wu, and Yiming Yang. Analogical inference for multi-relational embeddings.
> In ICML, volume 70, pp. 2168–2178, 2017.

---

> ### Author Response · Authors · 2022-11-24
> **Further discussion with Reviewer VDJN (denoted as R1)**
>
> Dear reviewer VDJN:
>
> We thank you for the precious review time and valuable comments. We have provided corresponding responses and results, which we believe have covered your concerns. We hope to further discuss with you whether or not your concerns have been addressed. Please let us know if you still have any unclear parts of our work.
>
> Best,
>
> Paper 3461 Authors.

---

> ### Author Response · Authors · 2022-12-10
> **Further Response to Reviewer VDJN (denoted as R1)**
>
> Thank you for your further concern and we apologize for not responding in a timely manner as we did not notice that you had added further questions to your original review.
>
> >R1-Q6: I would suggest to extend this experimental result to all the datasets, and incorporate them into the main body of the paper.
>
> Thanks for your suggestion, and we will put the following results to the main body in the final version. The following table is the ablation based on UniBi-O(2).
>
> |Constraints|WN18RR|FB15k-237|YAGO3-10-DR|
> |-|-|-|-|
> | Entity Norm + Spectral Radius Norm| 0.488 | 0.370 | 0.247 |
> |Entity Norm + Orthogonalization | 0.471 | 0.335 | 0.230 |
>
>
> >R1-Q7: Improvements are marginal.
>
> Thanks for your question, this is a similar question to R2-Q2 and R3-Q1, so in order to avoid a too lengthy response, we will only give the conclusion here, and you can refer to our previous responses for details. On the one hand, the performances on these datasets are reaching
> a glass ceiling, so the performances are hard to improve significantly. On the other hand, our main motivation is modeling the identity property theoretically rather than achieving much improvement practically, so the marginal improvements are acceptable.
>
> >R1-Q8:  By comparing RESCAL with UniBi, it does not seem that the proper normalization, as proposed in this paper, is the most important (or the only key) factor that affect the effectiveness.
>
> Thanks for your concern, but we just claim that proper normalization is useful rather than it is **the most important** or only key factor. Besides, as demonstrated in Fig. 9(a) and 9(b), we notice that RESCAL needs significantly more time and space, and we consider a comparable result between the two models to be acceptable.

---

### Author Response · Authors · 2022-11-24
**General Responses to Reviewers and ACs**

Dear Reviewers and ACs:

We sincerely appreciate all the reviews. They provide positive and high-quality comments on our paper with a lot of constructive feedback.

We have updated our draft to incorporate the insightful suggestions of the reviewers:

1. According to Reviewer 1 and Reviewer 4's question (R1-Q2, R1-Q3, R1-Q4, R4-Q1), we adjust our motivation and Section 4.4.1 to explain why modeling identity uniquely helps improve performance and emphasize the difference between our method and previous ones in terms of preventing ineffective learning with negligible cost.

2. According to Reviewer 2 and Reviewer 3's question (R2-Q2, R3-Q1), we add a potentially more powerful regularization term in Appendix I.

3. According to Reviewer 2's question (R2-Q1) and Reviewer 4's suggestion (R4-Q3), we add a more straightforward case to explain the idea of group and put the formal definition to Appendix D.

In the final version, we will improve other minor points of Reviewer 1, Reviewer 2, Reviewer 3, and Reviewer 4. Thank you all for the valuable suggestions.

Thanks,

Paper 3461 Authors.

---

### Author Response · Authors · 2022-12-10
**Our Responses to the Most Important Concerns of Reviewers**

Dear Reviewers and ACs:

We sincerely thank all reviewers for their efforts and Reviewers and ACs for their time. To facilitate potential communication with each other, here we summarize the key idea of our responses to the most important concerns of reviewers.

1) Modeling identity is not only interesting but also meaningful, with all 4 reviewers agreeing with the former and two doubting the latter (R3-Q2, R4-Q4). We believe that in theory, it is useful in three ways. a) Revise the existing framework and help to propose a proper solution. b) Understand the similarities and difference between bilinear and distance based models and help to propose an elegant unified model. c) Potentially useful for bilinear based models in other areas like knowledge graph alignment,  since alignment can be treated as finding the identity relation across different KGs. In a word, we believe that modeling identity per se is a strong motivation.

2) The performance improvements are reasonable (Response to R1-Q2, R4-Q1, and both Reviewer 1 and Reviewer 4 accept our explanation.). This is because our method prevents ineffective
learning with the least constraints. In addition, we also explain why it is the least (Response to R4-Q5) and demonstrate that the performance will drop if we add more constraints (Response to R1-Q7).

3) The marginal improvements are acceptable (Response to R1-Q7, R2-Q2, R3-Q1). On the one hand, the performance on these datasets is reaching a glass ceiling, so the performance is hard to improve significantly. On the other hand, our main motivation is modeling the identity property theoretically rather than achieving much improvement practically, and we didn't do what we could have done to try to improve performance even more, like searching for more hyperparameters or using more complex versions like UniBi-O(4), UniBi-O(n).

Finally, we thank again for the effort of Reviewers and ACs in processing this paper.

Thanks,

Paper 3461 Authors.

---

### Decision · Program_Chairs · 2023-01-20

**Decision:**

Reject

**Justification For Why Not Higher Score:**

The paper generally proposes an interesting analysis and achieves good performance. It was a very close decision not to accept the paper. The main reasons for the decision is that the novelty and the necessity of the central technique discussed in the paper was doubted. There was also the concern that the evaluation scores are not significantly better than previous work, but this was not used as major decision point for making the decision.

**Justification For Why Not Lower Score:**

N/A

**Metareview: Summary, Strengths And Weaknesses:**

The paper analyses the role of an identity relation in knowledge graphs for knowledge graph embedding methods, i.e. a relation that is reflexive and does not relate entities that are not identical. Based on observations for an identity relation, the authors propose a bilinear model for knowledge graph representation learning which shows good performance on benchmarks.

The reviewers identified the following strengths and weaknesses.

Strengths:
- Interesting analysis of identity from a theoretical point of view
- Experimental results are positive (performance improvements are not small, but this may be due to the number of KGE models published over the past few years with performance hitting a ceiling as well as this aspect not being the core aspect of the paper)
- Extensive ablation studies

Weaknesses:
- The paper essentially proposes a normalisation technique which in itself is only somewhat novel
- Paper is judged to be hard to follow by some reviewers
- Doubts by reviewers on the necessity to model identify relations in the sense that better models might by themselves model identity better which makes it unnecessary to model identity explicitly

**Summary Of Ac-Reviewer Meeting:**

Due to AC and reviewer timezone differences, we could not find a common time slot for a virtual meeting (even though we tried), but instead exchanged messages and made sure that the reviews acknowledge the author's response. Some concerns around reproducibility and motivation of the work could be resolved. However, reviewers saw no major reason to change (in particular the consideration to increase) review scores.